

**Low levels of nitryl chloride: Nocturnal nitrogen oxides in**
**the Lower Fraser Valley of British Columbia**
**Hans D. Osthoff[1], Charles A. Odame-Ankrah[1], Youssef M. Taha[1],**
**Travis W. Tokarek[1], Corinne L. Schiller[2], Donna Haga[3], Keith Jones[2], and**
**Roxanne Vingarzan[2]**
[1] {Department of Chemistry, University of Calgary, Calgary, Alberta T2N 1N4,
Canada}
[2] {Applied Science Division, Prediction and Services West, Meteorological Service of
Canada, Environment and Climate Change Canada, Vancouver, British Columbia V6C
3S5, Canada}
[3] {British Columbia Ministry of Environment and Climate Change Strategy,
Cranbrook, British Columbia V1C 7G5, Canada}
Correspondence to: H. D. Osthoff (hosthoff@ucalgary.ca)
**Abstract**
The nocturnal nitrogen oxides, which include the nitrate radical ($NO_3$), dinitrogen pentoxide
($N_2O_5$), and its uptake product on chloride containing aerosol, nitryl chloride ($ClNO_2$), can have
profound impacts on the lifetime of $NO_x$ (= $NO + NO_2$), radical budgets, and next-day
photochemical ozone ($O_3$) production, yet their abundances and chemistry are only sparsely
constrained by ambient air measurements.
Here, we present a measurement data set collected at a routine monitoring site near the
Abbotsford International Airport (YXX) located approximately 30 km from the Pacific Ocean
in the Lower Fraser Valley (LFV) on the West coast of British Columbia. Measurements were
made from July 20 to August 4, 2012, and included mixing ratios of $ClNO_2$, $N_2O_5$, NO, $NO_2$,
total odd nitrogen ($NO_y$), $O_3$, photolysis frequencies, and size distribution and composition of
non-refractory submicron aerosol.
At night, $O_3$ was rapidly and often completely removed by dry deposition and by titration with
NO of anthropogenic origin and unsaturated biogenic hydrocarbons in a shallow nocturnal



inversion surface layer. The low nocturnal $O_3$ mixing ratios and presence of strong chemical
sinks for $NO_3$ limited the extent of nocturnal nitrogen oxide chemistry at ground level.
Consequently, mixing ratios of $N_2O_5$ and $ClNO_2$ were low (<30 and <100 parts-per-trillion by
volume (pptv) and median nocturnal peak values of 7.8 pptv and 7.9 pptv, respectively). Mixing
ratios of $ClNO_2$ frequently peaked 1 - 2 hours after sunrise rationalized by more efficient
formation of $ClNO_2$ in the nocturnal residual layer aloft than at the surface and the breakup of
the nocturnal boundary layer structure in the morning. When quantifiable, production of $ClNO_2$
from $N_2O_5$ was efficient and likely occurred predominantly on unquantified supermicron sized
or refractory sea salt derived aerosol. After sunrise, production of Cl radicals from photolysis
of $ClNO_2$ was negligible compared to production of OH from the reaction of $O(^1D) + H_2O$
except for a short period after sunrise.
**Keywords**
Lower Fraser Valley, $ClNO_2$, surface measurements, nocturnal residual layer, $ClNO_2$ morning
peak, vertical entrainment



## 1 Introduction

The Lower Fraser Valley (LFV) is prone to episodes of poor air quality, in part because of its
geography which facilitates stagnation periods and accumulation of airborne pollutants through
processes such as the Wake-Induced Stagnation Effect (Brook et al., 2004), and also because
of continued growth of human population and associated emissions from urban, suburban,
agricultural and marine sources. Of special concern have been repeated exceedances of the
Canada-Wide Standard and, as of 2012, the Canadian Ambient Air Quality Standards (CAAQS)
for fine particulate matter ($PM_{2.5}$) and ozone ($O_3$) at Chilliwack and Hope, located in the eastern
part of the LFV downwind of Vancouver (Ainslie et al., 2013). These exceedances have
occurred in spite of ongoing declines in emissions of both nitrogen oxides ($NO_x = NO + NO_2$)
and volatile organic compounds (VOCs) resulting from the introduction of new vehicle
standards and (now discontinued) local vehicle emission testing programs (Ainslie et al., 2013).
Previous large-scale studies in the LFV such as Pacific 1993 (Steyn et al., 1997), the Regional
Visibility Experimental Assessment in the Lower Fraser Valley (REVEAL) I and II (Pryor et
al., 1997; Pryor and Barthelmie, 2000) and Pacific 2001 (Vingarzan and Li, 2006) have added
important information regarding atmospheric processes leading to $O_3$ and aerosol formation
and visibility issues. However, the transformation of primary (e.g., $NO_x$, VOCs, $SO_2$, $NH_3$, etc.)
to secondary pollutants (i.e.., $O_3$ and fine particulate matter) is highly complex, and the
scientific understanding of these highly non-linear processes remains incomplete.
A complicating factor in the LFV is the interaction of anthropogenic emissions with marine
derived sea salt aerosol. While sea spray aerosol is a primary source of particle matter (PM)
and hence directly affects particle concentration and mass loadings (Pryor et al., 2008), there is
now considerable evidence from modeling (Knipping and Dabdub, 2003), laboratory (Raff et
al., 2009), and field studies (Tanaka et al., 2003; Osthoff et al., 2008) that "active chlorine"
species released from sea salt can negatively affect air quality and promote $O_3$ and secondary
aerosol formation in coastal regions.
In an analysis of 20 years of $O_3$ air quality data in the LFV region, *Ainslie and Steyn* (2007)
concluded that precursor buildup, prior to an exceedance day, plays an important role in the
spatial $O_3$ pattern on exceedance days. Secondary processes involving active chlorine produced
from the interaction of marine aerosol with anthropogenic pollution would fit this profile but
are not currently constrained by measurements.



One pathway to activate chlorine from sea salt is the reactive uptake of dinitrogen pentoxide
($N_2O_5$) on chloride containing aerosol to yield nitryl chloride ($ClNO_2$). $N_2O_5$ is formed from
the reversible reaction of nitrogen dioxide ($NO_2$) with the photo-labile nitrate radical ($NO_3$),
which in turn is formed from reaction of $NO_2$ with $O_3$.
$$NO_2 + O_3 \rightarrow NO_3 + O_2 \qquad (1)$$
$$NO_3 + NO_2 \rightleftharpoons N_2O_5 \qquad (2)$$
During daytime, $NO_3$ (and, indirectly, $N_2O_5$) is removed primarily via its reaction with NO
(which is generated from $NO_2$ photolysis and directly emitted, for example, by automobiles)
and by $NO_3$ photolysis (Wayne et al., 1991).
$$NO_3 + NO \rightarrow 2NO_2 \qquad (3)$$
$$NO_3 + h\nu \rightarrow 0.9NO_2 + 0.1NO \qquad (4)$$
The heterogeneous hydrolysis of $N_2O_5$ to nitric acid ($HNO_3$) is an important nocturnal $NO_x$ and
odd oxygen ($O_x = NO_2 + O_3$) removal pathway (Chang et al., 2011; Brown et al., 2006a). On
chloride containing aerosol, however, uptake of $N_2O_5$ yields up to a stoichiometric amount of
$ClNO_2$ (Behnke et al., 1997; Finlayson-Pitts et al., 1989):
$$N_2O_5 + H_2O(het) \rightarrow (2\text{-}\phi)HNO_3(het) + \phi ClNO_2, \ 0 \leq \phi \leq 1 \qquad (5)$$
The $ClNO_2$ yield, $\phi$, is primarily a function of aerosol chloride and water content (Behnke et
al., 1997; Bertram and Thornton, 2009; Roberts et al., 2009; Ryder et al., 2014; Ryder et al.,
2015b; Ryder et al., 2015a). Formation of $ClNO_2$ impacts air quality in the following ways:
Since $ClNO_2$ is long-lived at night (Osthoff et al., 2008), its primary fate is photo-dissociation
(to Cl and $NO_2$) in the morning hours after sunrise.
$$ClNO_2 + h\nu \rightarrow NO_2 + Cl \qquad (6)$$
This reaction increases the morning abundance of $O_x$, leading to greater net photochemical $O_3$
production throughout the day. The other photo fragment, the Cl atom, is highly reactive
towards hydrocarbons and will initiate radical chain reactions that produce $O_3$ and secondary
aerosol (Behnke et al., 1997; Young et al., 2014). The fate and impact of $ClNO_2$ is thus similar
to that of nitrous acid (HONO), which also accumulates during the night and photodissociates
in the morning to release NO and the hydroxyl radical (OH) that go on to produce $O_3$ (Alicke
et al., 2003).



Data collected during the 2006 Texas Air Quality Study – Gulf of Mexico Atmospheric
Composition and Climate Study (TEXAQS-GOMACCS) have shown that $ClNO_2$ production
is efficient in the nocturnal polluted marine boundary layer even on primarily non-sea salt
aerosol surfaces. As a result, up to 15% of total odd nitrogen ($NO_y$) was present in the form
$ClNO_2$ at night (Osthoff et al., 2008). The high efficiency of $ClNO_2$ formation on aerosol of
medium-to-low total chloride content has been confirmed by several laboratory investigations
(Bertram and Thornton, 2009; Raff et al., 2009; Roberts et al., 2009) and direct measurements
of $N_2O_5$ uptake on ambient particles (Riedel et al., 2012b). Some ambiguity remains as to the
detailed mechanism of the reaction, but there is agreement that acid displacement of HCl from
supermicron (predominantly sea salt aerosol) to submicron (predominantly non-sea salt
aerosol) is a key step in the efficient production of $ClNO_2$. These results suggested that this
chemistry is active anywhere where pollution in the form of $NO_x$ and $O_3$ comes in contact with
marine air, including the LFV.
However, while the yield of $ClNO_2$ in reaction (5) is high in polluted coastal regions, the $ClNO_2$
yield relative to the amount of $NO_3$ produced from reaction (1) cannot be easily predicted
because $NO_3$ is consumed by reactions with VOCs, e.g., biogenic VOCs such as isoprene and
monoterpenes as well as aldehydes, and dimethyl sulfide (Wayne et al., 1991).
$$NO_3 + VOC \rightarrow products \tag{7}$$
Previous studies in the LFV have shown high biogenic VOC concentrations (Biesenthal et al.,
1997; Gurren et al., 1998; Drewitt et al., 1998) yet there was active nighttime nitrogen oxide
chemistry. During the Pacific 2001 study, measurements of the mixing ratios of NO, $NO_2$,
peroxyacetic nitric anhydride ($CH_3C(O)O_2NO_2$, PAN), HONO, $HNO_3$, and $NO_y$ at three ground
sites in the LFV indicated deficits of up to 15% in the nocturnal $NO_y$ budget (Hayden et al.,
2004) attributable to unquantified species such as alkyl nitrates, $N_2O_5$, and $ClNO_2$. *McLaren*
and coworkers quantified mixing ratios of $NO_2$ and $NO_3$ by differential optical absorption
spectroscopy (DOAS) at the Sumas Eagle Ridge site (~250 m above the floor of the LFV) as
part of Pacific 2001 (McLaren et al., 2004) and off-shore on Saturna Island (Figure 1) in the
Strait of Georgia in 2005 (McLaren et al., 2010). The LFV data showed occasional episodes of
active nocturnal nitrogen oxide chemistry in the residual layer with $N_2O_5$ contributing up to 9%
of $NO_y$, while the Saturna Island data showed $NO_3$ mixing ratios of > 20 parts-per-trillion by
volume ($10^{-12}$, pptv) every night of measurement. *McLaren et al*. estimated that between 0.3
and 1.9 ppbv of $ClNO_2$ would be produced under these conditions (McLaren et al., 2010).





Efficient formation of $ClNO_2$ would be consistent with the unidentified $O_3$ precursor proposed
by *Ainslie and Steyn* and is also a plausible explanation for part of the deficit in the $NO_y$ budget
observed by *Hayden et al.* (2004).
Another feature of the LFV are somewhat unusual diurnal profiles arising from the vertical
structure in pollutant concentrations. Measurements of $O_3$ and $NO_2$ using tethered balloons by
Pisano et al. (1997) during Pacific 93 at the Harris Road site (located ~38 km NW of Abbotsford
International Airport) revealed a highly stratified boundary layer with a shallow, 50 m deep
isothermal surface layer (also called a nocturnal boundary layer, or NBL) and low surface $O_3$
concentrations at night. Nocturnal loss of surface $O_3$ is known to occur by several pathways,
including dry deposition, titration with NO, and reaction with unsaturated biogenic
hydrocarbons (Neu et al., 1994; Kleinman et al., 1994; Trainer et al., 1987; Logan, 1989; Talbot
et al., 2005). Titration of $O_3$ with NO is readily quantified as the concentration of a product of
this reaction, $NO_2$, can be measured directly and conserves $O_x$.

150        $$O_3 + NO \rightarrow O_2 + NO_2 \tag{8}$$

Usually, the major nocturnal sink of $O_x$ is dry deposition of $O_3$ and $NO_2$ (Lin et al., 2010).
The balloon data also showed pools of $NO_2$ and $O_3$ in a ~100 m deep nocturnal residual layer
(NRL) located 200 to 350 m above ground. Following the break-up of the nocturnal layers in
the early morning, vertical down-mixing events of $O_3$ pollution were observed (McKendry et
al., 1997). In this process, pollutants are entrained into the growing mixed layer from the NRL,
i.e., the growing mixed layer in the hours after sunrise erodes the somewhat deeper NRL, and
pollutants are mixed to the surface (Neu et al., 1994; Kleinman et al., 1994).
In this manuscript, we present the first measurements of $ClNO_2$ and $N_2O_5$ mixing ratios in the
LFV. The data were collected at a surface site east of the Abbotsford International Airport
(International Air Transport Association (IATA) airport code YXX) located approximately
35 km from the Pacific Ocean from July 20 to August 5, 2012. Auxiliary measurements
included NO, $NO_2$, $NO_y$, $O_3$, photolysis frequencies, and submicron aerosol composition and
size distributions. An analysis of nocturnal nitrogen oxide chemistry including the formation of
$ClNO_2$ and its potential impact on nocturnal $O_3$ and $NO_2$ loss and radical budgets in the LFV
are presented.



## 2 Experimental

### 2.1 Location

The map shown in Figure 1 indicates the location of the study. Ambient air measurements were
conducted at the T45 routine monitoring site located to the east YXX at latitude 49.0212 (N)
and longitude -122.3267 (W) and ~60 m above sea level (ASL) and ~30 km from the Pacific
Ocean. A raspberry field was located immediately to the W between the end of the airport
runway and the measurement site. Nearby local sources included agricultural operations (such
as poultry farms) and emissions from motor vehicle traffic on secondary roads and highways.
YXX is located ~60 km ESE of the Vancouver International Airport (YVR) and the City of
Vancouver. Abbotsford is in the heart of the so-called "Lower Mainland", the low-lying region
stretching from Pacific Ocean at Vancouver to the NW and the Canada-USA border to the S
(north of Bellingham, BLI) to the eastern end of the Fraser Valley with a total population in
excess of 2,500,000.

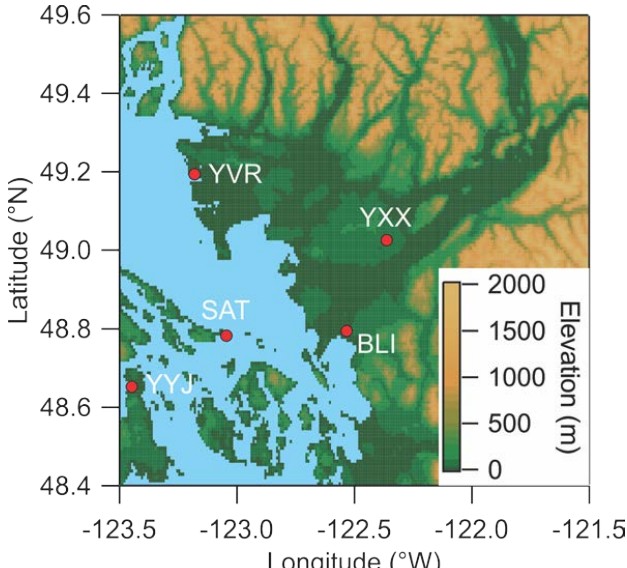

**Figure 1.** Map of the Lower Fraser Valley. YXX = Abbotsford International Airport (measurement location for this study). YVR = Vancouver Int'l Airport. YYJ = Victoria Int'l Airport. BLI = Bellingham Int'l Airport. SAT = Saturna Island.





**2.2  Measurement techniques**
The measurement techniques used for this study are listed in Table 1. Data were averaged to 5
min prior to presentation.
The instruments measuring $O_3$ and nitrogen oxides were housed in an air-conditioned trailer
and sampled from a common 0.635 cm (¼") outer diameter (o.d.) and 0.476 cm (3/16") inner
diameter (i.d.) Teflon™ inlet at a height of 4 m above ground; the setup is depicted in Figure 3
of *Tokarek et al.* (2014). A scroll pump whose flow rate was throttled using a 50 standard liters
per minute (slpm) capacity mass flow controller was connected to the end of the common inlet
to minimize the residence time of the sampled air and to reduce inlet "ageing", i.e.,
accumulation of aerosol on filters of individual instruments, whose inlets tapped into the main
inlet line at 90°. The total inlet flow was in the range of 18 to 20 slpm.
Measurements of aerosol composition and size distributions and of meteorological data were
made from the research trailer housing the routine measurements at the site. The Agilent VOC
measurements were made from a research trailer owned by ECCC.

2.2.1  Quantification of $ClNO_2$ by iodide chemical ionization mass spectrometry
Mixing ratios of $ClNO_2$ were quantified as iodide cluster ions at *m/z* 208 using the THS iodide
chemical ionization mass spectrometer (iCIMS) described by *Mielke et al.* (2011) and calibrated
using the scheme by *Thaler et al.* (2011). In this method, a gas stream containing $ClNO_2$ is
generated from reaction of $Cl_2$ (Praxair, 10 ppmv in $N_2$) with an aqueous slurry saturated with
$NaNO_2$ (Sigma-Aldrich):
$$Cl_2(g) + NO_2^-(aq) \rightleftharpoons ClNO_2(g) + Cl^-(aq) \tag{9}$$
This gas stream was periodically added to the main inlet with the aid of a normally-open 2-way
valve connected to a vacuum pump in a similar fashion as described earlier for $N_2O_5$ and PAN
(Tokarek et al., 2014; Odame-Ankrah and Osthoff, 2011). The $ClNO_2$ content of the calibration
gas stream was quantified by thermal dissociation cavity ring-down spectroscopy (TD-CRDS)
as described in section 2.2.2. In total, 31 calibrations for $ClNO_2$ were carried out, spread out
evenly over the measurement period. The iCIMS response factor at *m/z* 208 was (0.40±0.06)
Hz pptv$^{-1}$ (where the error represents the standard deviation of repeated calibrations),
normalized to $10^6$ counts of reagent ion at *m/z* 127. The $^{37}ClNO_2I^-$ ion at *m/z* 210 was also
monitored and found to be (0.298±0.004) times the signal at *m/z* 208 ($r^2 = 0.944$), slightly lower





than Standard Mean Ocean Chloride $^{37}Cl$ mole fraction in sea water of ~0.319 (Wieser and
Berglund, 2009) and our previously observed ratios of 0.315±0.003 in Calgary (Mielke et al.,
2011) and 0.3065±0.0002 in Pasadena (Mielke et al., 2013). The reason(s) for these differences
are unclear but may be a result of fractionation processes (Koehler and Wassenaar, 2010; Volpe
et al., 1998), a topic outside the scope of this manuscript.
The iCIMS was also used to quantify mixing ratios of PAN at $m/z$ 59 and PPN at $m/z$ 73 (Slusher
et al., 2004; Mielke et al., 2011; Mielke and Osthoff, 2012). For this reason, part of the
instrument's inlet prior to the ion-molecule reaction region was heated to 190 °C to dissociate
PANs into their respective carboxylates. Further, the collisional dissociation chamber (CDC)
was operated in declustering mode (-22.7 V) to break up ion clusters. Calibrations and matrix
effect correction procedures and a time series of the PAN and PPN data were presented by
*Tokarek et al.* (2014).

## 2.2.2  Quantification of $NO_2$ and $N_2O_5$ by cavity ring-down spectroscopy
The CRDS used in this work was an amalgamated version of two instruments described earlier
(Paul and Osthoff, 2010; Odame-Ankrah and Osthoff, 2011), called "Improved Detection
Instrument for Nitrogen Oxide Species" (iDinos) (Odame-Ankrah, 2015). A schematic of the
optical layout is shown in Figure 2. The optical bread board, instrument frame, electronic and
data acquisition components were as described by *Paul and Osthoff* (2010). The new instrument
was set up with up to six parallel detection channels: four 405 nm "blue" diode laser CRDS
cells for quantification at $NO_2$ via its absorption at 405 nm with a distance between the pairs of
high-reflectivity (HR) mirrors (Advanced Thin Films) of 112.5 cm, of which 92.0 cm were
filled with sample air, and two newly constructed 662 nm "red" diode laser CRDS cells for
quantification at $NO_3$ via its absorption at 662 nm with a distance between the HR mirrors (Los
Gatos) of 93.0 cm of which 73.0 cm were filled with sample air. Light exiting the far ends of
the CRDS cells was collected using fixed-focus collimating lenses and multimode optical fibers
(Thorlabs) connected to photomultiplier tubes (PMT, Hamamatsu H9433-03MOD) with 10
MHz bandwidth. Bandpass filters (Thorlabs FB405-10 and FB660-10) were placed between
the PMTs and the end of the optical fibers.
The two laser diodes were simultaneously square-wave modulated by a function generator (SRS
DS335). The PMT voltages were digitized using an 8-channel 14-bit data acquisition card
(National Instruments PCI-6133; 2.5 MS s$^{-1}$ simultaneous sampling sample rate) connected to



a laptop computer via a PCMCIA-to-PCI expansion unit (Magma CB4DRQ) and controlled by
software written in LABVIEW™ (National Instruments).
Ring-down time constants ($\tau$) were determined from a linear fit to the logarithm of the digitized
PMT voltage as described by *Brown et al.* (2002) immediately after acquisition of the ring-
down traces (which were co-added to a user-selectable averaging time prior to the fit). The
fitting algorithm requires the subtraction of the PMT voltage offset prior to taking the logarithm;
this offset was measured between ring-down events after the signal had returned to baseline,
which limited the repetition rate of the diode lasers and the number of traces averaged per
second to a frequency of 300 Hz.

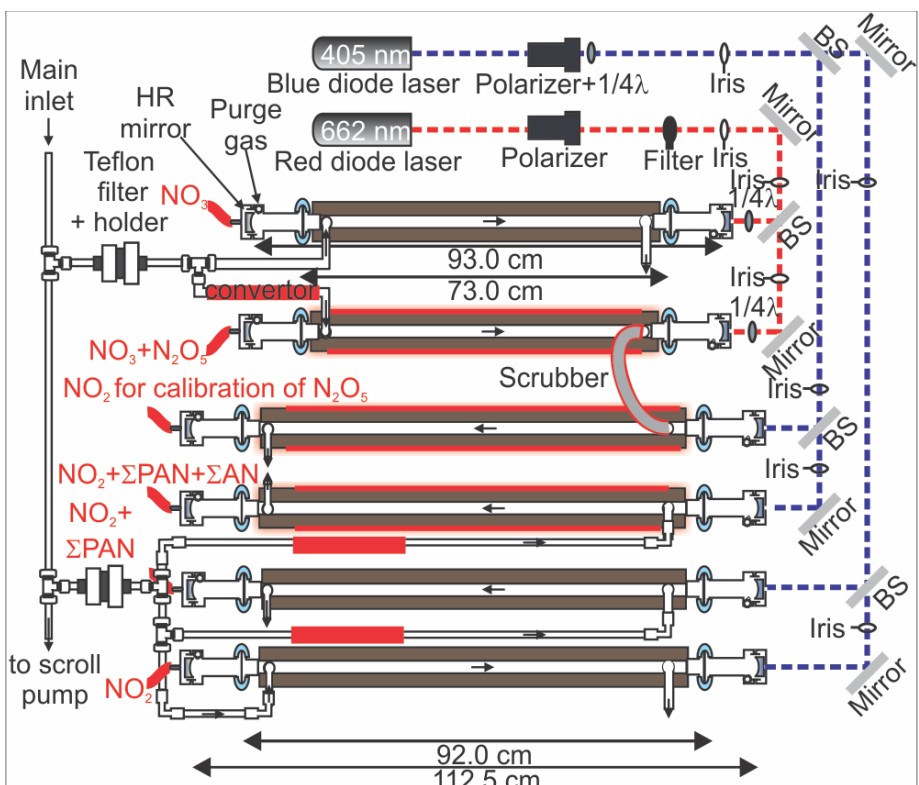

**Figure 2**. Optical layout of the cavity ring-down spectrometer. ¼$\lambda$ = quarter wave plate. BS =
beam splitter. HR mirror = high reflectivity mirror. Drawing is not to scale.



Ring-down time constants in the absence of the target absorber ($\tau_0$) were determined by
flooding the inlet (each once per hour) with ultra-pure, or "zero", air (Praxair) for the 405 nm
channels and by titration with NO for the 662 nm channel (Brown et al., 2001; Simpson, 2003)
Typical values of $\tau_0$ were in the range of 63 to 67 μs and between 198 and 210 μs for the blue
and red channels, respectively. The baseline precision (i.e., standard deviation, $\sigma$) of the $NO_2$
and $NO_3$ measurements were ±80 pptv and ±3 pptv (1 s data), respectively. For the $NO_3$
channels, additional noise was introduced by variable background absorption of $NO_2$, $O_3$, and
water vapor which produce small, spurious structure in the 662 nm absorption signal (Dubé et
al., 2006) and were not tracked well by the interpolation of the baseline from the hourly $\tau_0$
determinations.
During the Abbotsford campaign, only five (four blue and one red) CRDS channels were
operated because of delays in the fabrication of the final set of CRDS mirror holders. The
662 nm CRDS cell sampled from a Teflon™ inlet heated to 130 °C for quantification of $NO_3$
plus the $NO_3$ generated from thermal dissociation $N_2O_5$ (Brown et al., 2001; Simpson, 2003;
Dubé et al., 2006). Under the high-$NO_x$ conditions of this study, equilibrium (2) was sufficiently
far to the right (see section 3.3) such that $[NO_3] + [N_2O_5] \approx [N_2O_5]$, i.e., the concentration
measured could be equated with $[N_2O_5]$ without introducing a large error (i.e., <5%). The four
405 nm CRDS cells were operated as follows: The first sampled from an ambient temperature
inlet and was used to quantify $NO_2$. The second sampled from a quartz inlet heated to 250 °C
and was used to quantify $NO_2$ plus total peroxyacyl nitrate ($\Sigma PAN$) (Paul et al., 2009; Paul and
Osthoff, 2010). Data from this channel will be presented in a future manuscript. The third was
operated with a quartz inlet heated to 450 °C to enable $ClNO_2$ calibrations (Thaler et al., 2011).
Quantification of total alkyl nitrates ($\Sigma AN$) in ambient air was not attempted because of the
high $NO_x$ levels and resulting large subtraction errors (Thieser et al., 2016). The fourth 405 nm
CRDS cell was connected with polycarbonate tubing (⅜" o.d. and ¼" i.d.) in series to the
662 nm channel and was used to calibrate the response of the $N_2O_5$ channel, which is a function
of the transmission efficiency of $N_2O_5$ through the inlet and the overlap of the diode laser
spectrum with the $NO_3$ absorption line (Odame-Ankrah and Osthoff, 2011). The role of the
polycarbonate tube was to scrub $NO_3$ exiting the $N_2O_5$ channel, allowing detection of only the
$NO_2$ generated from thermal dissociation of $N_2O_5$ and to prevent recombination of $NO_3$ and
$NO_2$ in the blue calibration channel (Wagner et al., 2011).





$N_2O_5$ was generated in situ by adding an excess of $O_3$ (generated by passing $O_2$ past a 254 nm
Hg lamp) to nitric oxide (NO) in a 0.635 cm (¼") o.d. and 0.476 cm (3/16") i.d. Teflon™
calibration line and allowed to equilibrate (i.e., until the output was constant) offline before
being switched inline on demand. The $N_2O_5$ response varied between 65% and 100% depending
on inlet "age"; the Teflon™ inlet and aerosol inlet filter were changed every 2 – 3 days. The
accuracy of the $NO_2$ and $N_2O_5$ data were ±10% and ±25%, respectively, driven mainly by the
systematic uncertainty of the $NO_2$ absorption cross-section and of the $N_2O_5$ inlet transmission
efficiency (Odame-Ankrah, 2015).

### 303  2.2.3  Measurements of $O_3$, NO and $NO_y$,

Mixing ratios of $O_3$ were monitored by UV absorption in a commercial instrument (Thermo 49)
and were accurate within ±2% and ±1 ppbv. An $NO$-$O_3$ chemiluminescence instrument
(Thermo 42i) was used to monitor mixing ratios of NO and $NO_y$, which was reduced to NO in
a Mo converter heated to ~320 °C. This instrument sampled from the main inlet via a Teflon™
filter and filter holder and was calibrated daily against CRDS as described by *Tokarek et al.*
(2014). The slope uncertainty for each multipoint calibration was ±15%. Interpolation between
calibration runs gave an overall uncertainty of ±30%. The NO zero offset uncertainty (needed
for calculating the $NO_3$ loss rate with respect to reaction with NO) was ±10 pptv.

### 313  2.2.4  VOC measurements

Volatile organic compounds were monitored with a commercial gas chromatograph - mass
spectrometer (GC-MS; Agilent model 7890A and 5975C) equipped with an FID detector and a
Markes Unity 2 pre-concentrator with an ozone precursor trap cooled to -25 °C.
In a typical sampling sequence, a 500 mL air sample was collected at a flow rate of 25 mL
min$^{-1}$, taken from the center flow of a 1.27 cm (½") stainless steel inlet line which was
continuously sampling ambient air at 5 L min$^{-1}$. The sampled air flowed through a 0.318 cm
(1/8") stainless steel line and particles were removed using a 1 µm pore size fritted filter. Once
500 mL of air were collected, the pre-concentrator was flushed with helium to remove air while
awaiting injection.  At the start of a GC run, the sample in the pre-concentrator was flash heated
to 300 °C and held for 3 min. The sample was separated on 2 columns with the entire sample





going through the Agilent VRX column with a Dean switch directing the first gases emitted to
a second GasPro column and then to the FID detector (~<C4) while the heavier compounds
were detected using the MS detector in scan mode.
The cycle time for the GC analysis was 1 hour with the sample being collected during the
previous runs analyses. The 20 min sample was taken at the start of a 1 hour time period.
Due to the low temperature of the trap, the air was dried using a trap at -30 °C. The trap was
heated and dried between each sample and reconditioned for 10 min prior to sample collection.
All sample lines were stainless steel with a Restek Sulfinert™ coating to minimize sample loss
on the lines. Calibrations were performed once per day for 105 species using a 100 ppbv U.S.
Environmental Protection Agency (EPA) photochemical assessment monitoring system
(PAMS) and a 100 ppb EPA air method, toxic organics – 15 (TO15) standard tanks (Linde
Specialty Gases) at an approximately concentration of 2 ppbv. The terpenes were semi-
quantitatively measured as a calibration source was not available at the time and only the
changes in concentration strength with time of day used. The accuracy of the measurements
varied depending on the species but was better than ±30% throughout. Peaks were manually
reintegrated using Chemstation software from Agilent. Table S-1 summarizes the VOCs
quantified.

**2.3  Aerosol measurements**
The chemical composition of non-refractory submicron particulate matter was monitored using
an Aerosol Chemical Speciation Monitor (ACSM, Aerodyne), which reported concentrations
of $NO_3^-$, $SO_4^{2-}$, $Cl^-$, $NH_4^+$, and total organics. A general description of this instrument designed
for routine monitoring has been given by Ng et al. (2011).
Submicron aerosol size distributions were quantified by a scanning mobility particle sizer
(SMPS, TSI 3034). This instrument measured aerosol particles in the range from 10 to 487 nm
using 54 size channels (32 channels per decade). Both of these instruments were housed in a
trailer operated by Metro Vancouver. The ACSM and the SMPS sampled air off a shared
stainless steel inlet that had a total flow of 5 L min$^{-1}$ and contained a $PM_{2.5}$ sharpcut filter at the
inlet and was operated at ambient relative humidity.





## 2.4  Photolysis frequencies
Photolysis frequencies were determined by solar actinic flux spectroradiometry (Hofzumahaus
et al., 1999) using a commercial radiometer with $2\pi$ receptor optics and photo diode array
(PDA) detector (Metcon; 512 pixels, wavelength range 285 nm - 690 nm) calibrated by the
manufacturer. The spectrometer was mounted facing up (zenith view) and hence measured the
down-dwelling radiation. On several days, the spectrometer was inverted hourly to determine
the up-dwelling radiation, which was added to the down-dwelling flux. Photolysis frequencies
including $j(NO_3)$, $j(NO_2)$, $j(O^1D)$, and $j(ClNO_2)$ were calculated using reference spectra and
quantum yields from (Sander et al., 2010) and (Ghosh et al., 2012). Table 2 gives the ratio of
observed up-dwelling to down-dwelling for selected photolysis frequencies. For August 3 (a
cloud-free day), the measurements were compared to (hourly) predictions with the online
"Tropospheric Ultraviolet and Visible (TUV) Radiation Model" V5.0 (Madronich and Flocke,
1997); with default settings, the model reproduced the measured $j(NO_2)$ and $j(O^1D)$ quite well:
a scatter plot of observed against TUV rate constants had correlation coefficients (r) of 0.997
and 0.998, slopes of 1.06±0.02 and 1.10±0.02, and offsets of $(3\pm1)\times10^{-4}$ $s^{-1}$ and $(5\pm3)\times10^{-7}$ $s^{-1}$.
## 2.5  Box model simulations of the nocturnal $O_3$ and $O_x$ loss in the NBL
A box model was set up to reconcile the median nocturnal decays of $O_3$ and $O_x$. These
simulations are intended as back-of-the-envelope type estimates of major processes only since
an accurate description of the nocturnal boundary layer chemistry would require modeling of
horizontal and vertical transport, i.e., altitude-resolved information not available in this study
(Geyer and Stutz, 2004). The model's assumptions are a well-mixed NBL that is decoupled
from the NRL above it as observed by earlier balloon vertical profiling (Pisano et al., 1997), $O_3$
and $NO_2$ dry deposition velocities of $v_d(O_3) = 0.2$ cm $s^{-1}$ and $v_d(NO_2) = \alpha \times v_d(O_3)$ with $\alpha$=0.65
(Lin et al., 2010), and negligible chemical $O_3$ and $O_x$ losses other than titration of $O_3$ by NO
(reaction 8) and by reaction with a generic biogenic hydrocarbon (assumed to react with $O_3$
with a rate coefficient of $5\times10^{-11}$ $cm^3$ $molec.^{-1}$ $s^{-1}$, i.e., the rate coefficient for reaction of $\alpha$-
pinene with $O_3$ (Seinfeld and Pandis, 2006)). Simulations were initiated with the median $NO_2$
and $O_3$ concentrations observed at sunset.



## 3 Results

### 3.1 Overview of data set

#### 3.1.1 Meteorology

A time series of local wind direction and speed are displayed in Figure 3D. During the two-week long measurement period, the air flow to the site was from the Pacific Ocean to the SW and WSW with a moderate wind speed of 8.7 km hr$^{-1}$ (median value). On most nights, local wind speeds were calm, i.e., $< 5$ km hr$^{-1}$ (median speed 3.6 km hr$^{-1}$) and from variable directions, though predominantly from the W and N. The two exceptions were the nights of July 22/23 and August 1/2 when stronger winds ($> 5$ km hr$^{-1}$) from the W and SW persisted. These nights saw relatively high ClNO$_2$ mixing ratios (see section 3.1.4).

The air temperatures were quite mild and ranged from a minimum of 11.0 °C to a maximum of 31.9 °C. The warm temperatures shifted equilibrium K$_2$ from N$_2$O$_5$ towards NO$_3$ and NO$_2$ (further analyzed in section 3.2.2). At night, temperatures frequently dropped to the dew point, resulting in occasional fog formation (shown as grey rectangles in Figure 3D), sometimes after sunrise. Fog droplets are strong sinks for N$_2$O$_5$ (Osthoff et al., 2006). In total, the impact of fog was minor, affecting 5% of the data. In addition, there were two periods with precipitation: The first occurred intermittently on July 20 until the morning of July 21. The second rainfall event was a 24-hour period from mid-day July 22 to the afternoon of July 23 (shown as blue dots in Figure 3D). July 23 also exhibited the highest wind speeds of the campaign (Figure 3C) and lowest daytime photolysis frequencies. The time series of j(ClNO$_2$) is shown as a representative example in Figure 3A. The photolysis data indicates that it was sunny on 6 days (July 25, 26, 29, Aug 1, 4 and 5) and that the remaining days had variable cloud cover, consistent with hourly meteorological logs that showed 10% of the measurement period affected by precipitation.






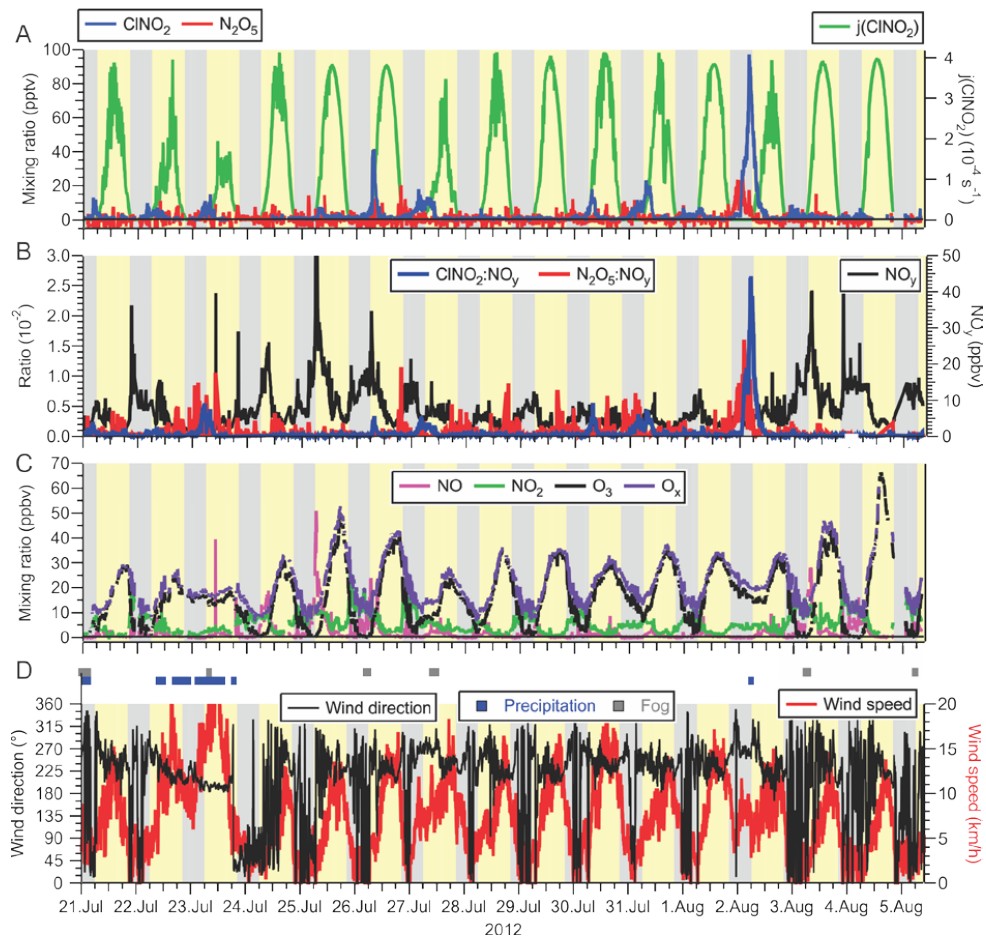


**Figure 3**. (**A**) Time series of $N_2O_5$ and $ClNO_2$ mixing ratios (left axis) and $ClNO_2$ photolysis
frequency (right axis) observed at T45 near the Abbotsford International Airport. (**B**) Time
series of the ratios of $ClNO_2$ and $N_2O_5$ to $NO_y$ (left axis) and of $NO_y$ (right axis). (**C**) Time
series of NO, $NO_2$, $O_3$, and $O_x$ (= $NO_2$ + $O_3$) mixing ratios. (**D**) Time series of local wind
direction (left axis) and speed (right axis). The blue and grey dots above the time series indicates
periods of precipitation (drizzle or rain) and fog, respectively, as identified in hourly
meteorological logs.




### 3.1.2  NO and $NO_2$

The rates of $N_2O_5$ and $ClNO_2$ formation depend on the rate of $NO_3$ production,

$P(NO_3)=k_1[NO_2][O_3]$ (analyzed further in section 3.2.2); therefore, it is informative to first
examine the mixing ratios of $NO_2$ and $O_3$ (see section 3.1.3). The time series of NO, $NO_2$, $O_3$,
and $O_x$ $(= O_3 + NO_2)$ mixing ratios are shown in Figure 3C, and their diurnal averages are shown
as $10^{th}$, $25^{th}$, $50^{th}$, $75^{th}$ and $90^{th}$ percentiles in Figures 4B and 4C.
The median NO and $NO_2$ mixing ratios for the entire campaign were 0.9 and 5.9 ppbv,
respectively. The average $NO_x/NO_y$ ratio for the entire campaign was 0.89 (data not shown).
These concentration levels are characteristic of an urban air mass impacted by relatively fresh
emissions from combustion engines in automobiles.
At night, mixing ratios of NO were generally lower than during the day though not negligible
(median 0.3 ppbv, Figure 4B) as NO was oxidized by $O_3$ to $NO_2$ (reaction 8) and was not
replenished by $NO_2$ photolysis. However, mixing ratios of NO increased throughout the night,
often coinciding with complete nocturnal removal of $O_3$ (see section 3.1.3), which indicates the
presence of nearby combustion sources of $NO_x$ (most likely automobile exhaust). The presence
of NO titrates $NO_3$ (via reaction 3) and effectively shut down $N_2O_5$ and $ClNO_2$ production for
most of the study: 68% of the measurement period had NO mixing ratios > 100 pptv and $NO_3$
lifetimes (with respect to its reaction with NO) of < 15 s. In contrast, $NO_2$ mixing ratios were
highest at night (median 7.3 ppbv), amplified further by $NO_x$ emissions that continued
throughout the night and likely by low nocturnal mixing heights (see discussion).
Mixing ratios of NO and $NO_x$ were highest in the morning hours. Concentration changes at this
time of day are difficult to interpret since the NBL breaks up during this time, resulting in
vertical mixing of air masses, photolabile species (e.g., $ClNO_2$, HONO, $N_2O_5$, etc.) that
accumulated overnight begin to photodissociate, and local emissions change with the onset of
rush hour.
In contrast to the morning increase in NO, an afternoon/early evening maximum in NO was
absent. This can be rationalized by a greater abundance of oxidants that oxidize NO to $NO_2$,
i.e., $O_3$ (see Figures 3 and 4) and organic peroxy radicals in the afternoon, a topic outside the
scope of this manuscript.



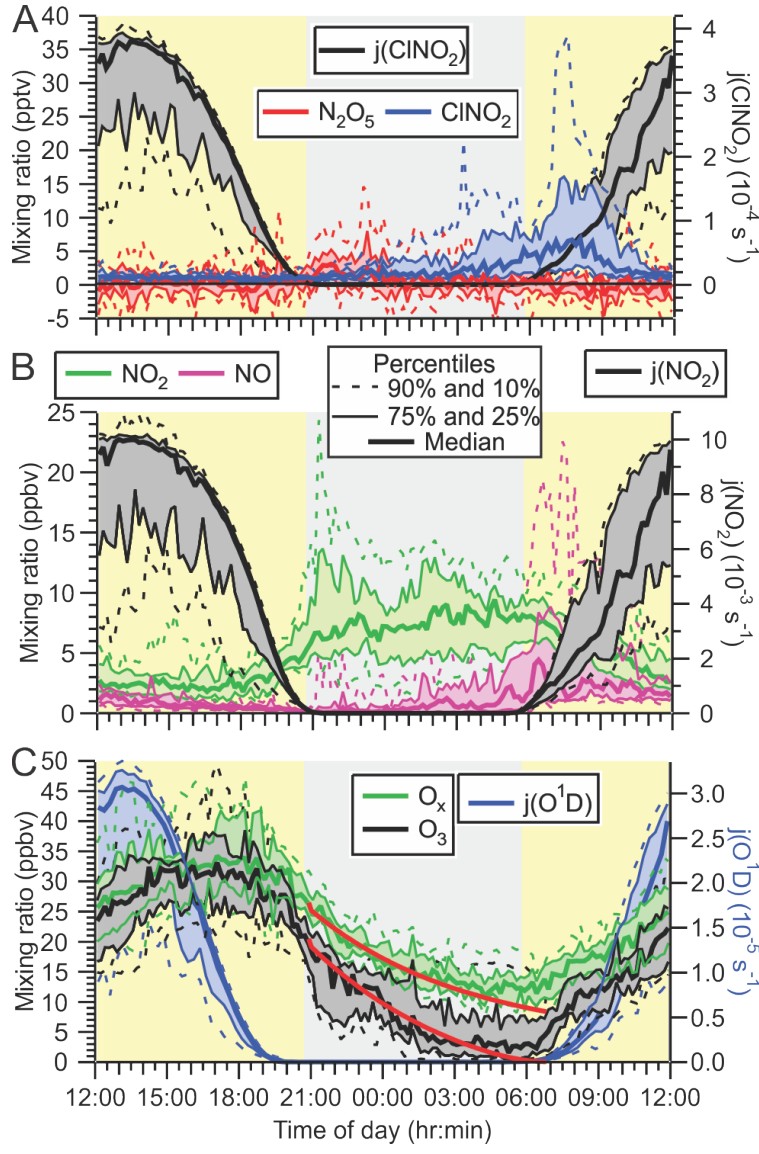


**Figure 4**. (**A**) Diurnal variation of $ClNO_2$ and $N_2O_5$ mixing ratios (left axis) and $ClNO_2$

photolysis frequencies (right axis). (**B**) Diurnal profiles of NO and $NO_2$ (left axis) and $NO_2$

photolysis frequency (right axis). (**C**) Diurnal profiles of $O_3$ and $O_x = O_3 + NO_2$ (left axis) and

$O_3 \rightarrow O(^1D)$ photolysis frequency (right axis). The superimposed lines shown in red are results

from a simple box model (see text).




### 3.1.3 $O_3$ and $O_x$

The time series of $O_3$ mixing ratios and its diurnal profile are shown in Figure 3C and 4C,
respectively. $O_3$ mixing ratios were small (average $\pm$ 1 standard deviation of 16±12 ppbv) and
peaked at ~17:00 in the afternoon. The highest concentrations were observed on August 4 from
13:55 to 15:30, when mixing ratios were 64.4±1.2 ppbv (the 8-hour running average was 52
ppbv). These levels were well below the CAAQS 8-hr standard of 63 ppbv and the 1 hour
National Ambient Air Quality Objective of 82 ppbv, smaller than the pre-2003 data analyzed
by *Ainslie and Steyn* (2007) and of similar magnitude as observed by a high-density monitoring
network in the region in 2012 (Bart et al., 2014).
A recurring feature of this data set was the rapid and often complete loss of $O_3$ at night (Figure
4C). This was accompanied by an increase in the $NO_2$ mixing ratios, though by less (+6 ppbv
on average) than the amount of $O_3$ that was lost (-26 ppbv on average), showing that NO to
$NO_2$ conversion (reaction 8) was a contributor, though minor (~25%) to the nocturnal $O_3$ loss.
The diurnal profile of $O_x$ was similar to that of $O_3$, in that the highest concentrations occurred
in the afternoon (at ~18:00) and a considerable fraction of $O_x$ was removed at night: At sunset,
a median amount of 26 ppbv of $O_x$ were present, which decreased to 12 ppbv at sunrise (Figure
4C). The pathways contributing to nocturnal $O_3$ and $O_x$ loss of are probed using box model
simulations in section 3.2.1.
There were two (out of 16 total) nights when $O_3$ was not completely removed: on July 22-23
and August 1-2, $O_3$ mixing ratios dropped from a daytime maxima of ~33 ppbv to non-zero
nocturnal minima of ~16 ppbv. On both of these nights, $ClNO_2$ and $N_2O_5$ mixing ratios were
elevated (Figure 3A), and the two largest $ClNO_2$ to $NO_y$ ratios were observed (Figure 3B). The
local wind speeds were > 6 km hr$^{-1}$, whereas on other nights, local winds were calmer (Figure
3C). The greater local wind speeds likely induced more turbulence and a higher vertical mixing
height.



### 3.1.4 $N_2O_5$ and $ClNO_2$

Time series of $ClNO_2$ and $N_2O_5$ mixing ratios and $ClNO_2$ photolysis frequencies are shown in Figure 3A. Mixing ratios of $ClNO_2$ and $N_2O_5$ were small (campaign averages at night of 4.0 pptv and 1.4 pptv, respectively). The mixing ratios peaked prior to sunrise at a median value of 7.9 and 7.8 pptv for $ClNO_2$ and $N_2O_5$, respectively. The highest mixing ratio of this campaign was 97 pptv for $ClNO_2$ and 23 pptv for $N_2O_5$, both observed on the night of August 1-2. This night was also the only time when nocturnal $ClNO_2$ mixing ratios exceeded 20 pptv and is analyzed in greater detail in section 3.2.3.

Consistent with their low mixing ratios, neither $ClNO_2$ nor $N_2O_5$ were significant components of $NO_y$ (Figure 3B): on average, they contributed 0.1% to the nocturnal $NO_y$ budget, though $NO_y$ mixing ratios were large (median 6.3 ppbv at night), typical for a site impacted by urban emissions. The only exception was the night of August 1-2, when $ClNO_2$ and $N_2O_5$ constituted 2.6% and 1.6% of $NO_y$, respectively, and $NO_y$ mixing ratios were 4.4 ppbv on average (Figure 3B).

The $ClNO_2$ and $N_2O_5$ mixing ratios are displayed as functions of time of day in Figure 4A. Before midnight local time, $N_2O_5$ mixing ratios were slightly larger (median value of 1.8 pptv on average) than those of $ClNO_2$ (median value of 1.4 pptv on average), whereas after midnight, $ClNO_2$ mixing ratios were larger than those of $N_2O_5$ (2.0 pptv vs. 0.6 pptv). The latter is consistent with observations at other ground sites, which generally showed higher concentrations of the longer-lived $ClNO_2$ prior to sunset (Thornton et al., 2010; Mielke et al., 2013). The higher $N_2O_5$ than $ClNO_2$ abundances at the beginning of the nights suggests that the $N_2O_5$ production rate at that time exceeded its ability to react heterogeneously and convert to $ClNO_2$, potentially due to a lack of available aerosol chloride or otherwise reduced $N_2O_5$ heterogeneous uptake parameters (Thornton et al., 2010).

Production of $ClNO_2$ from $N_2O_5$ uptake on aerosol ceases after sunrise because of the rapid removal of $N_2O_5$ and $NO_3$ as the latter is titrated by NO and destroyed by photolysis (reactions 3 and 4) (Wayne et al., 1991). In spite of this, $ClNO_2$ mixing ratios frequently (on 12 out of 15 measurement days) continued to increase after sunrise (Figures 3A and 4), peaking on average at ~07:45 in the morning approximately 2 hours after sunrise. The median mixing ratio at that time was 6.7 pptv larger than the median value of 5.3 pptv observed at sunrise. The most prominent example of this phenomenon occurred on the morning of July 26. For a two hour period leading up to sunrise, there was fog (virtually ensuring the absence of $N_2O_5$), and $ClNO_2$





mixing ratios were < 5 pptv. The fog then dissipated at sunrise. One hour later, $ClNO_2$ mixing
ratios increased to > 40 pptv. Similar events (though with more modest $ClNO_2$ increases) were
observed on the mornings of July 22, 23, 25, 27, 28, 30, 31, and Aug 1. Two of these (July 23
and 27) overlapped with brief fog events.
Qualitatively similar $ClNO_2$ morning peaks have been observed at other ground sites and were
rationalized by vertical mixing (Tham et al., 2016; Bannan et al., 2015; Faxon et al., 2015).
In the period after the $ClNO_2$ morning peak after ~09:00, $ClNO_2$ mixing ratios decreased,
coinciding with the increasing $ClNO_2$ photolysis rate. Box model simulations (see S.I.) indicate
that the decay of $ClNO_2$ (after 09:00) was consistent with its destruction by photolysis.
There were two exceptions: the mornings of July 27 and Aug 2, when the decay of $ClNO_2$
concentration occurred at a rate faster than its photolysis. On July 27, fog was not observed
until 8:00, at which time the $ClNO_2$ mixing ratio rapidly decreased because of dissolution and/or
an air mass shift to one with a different chemical history. On Aug 2, the campaign maximum
of 97 pptv was observed at 04:40 prior to sunrise, followed by a sharp decline. Hourly logs
indicated scattered showers at 06:00.

### 3.1.5 Aerosol size distribution and composition measurements

The time series of submicron surface area density ($S_A$) observed by the SMPS is shown in
Figure 5A. The aerosol loadings were modest: the average (median) surface area density was
128 (104) $\mu m^2\ cm^{-3}$ and ranged from extremes of 26 to 618 $\mu m^2\ cm^{-3}$. Shown on the right hand
side is the rate coefficient for heterogeneous uptake of $N_2O_5$, $k_{N_2O_5}$ calculated using equation

532    (10).

$$k_{N_2O_5} = \frac{1}{4}\gamma\bar{c}S_A \tag{10}$$
Here, $\gamma$ and $\bar{c}$ are the uptake probability and the mean molecular speed of $N_2O_5$, respectively
(Davidovits et al., 2006). For this calculation, a $\gamma$ value of 0.025 was assumed. The average (±1
standard deviation) of $k_{N_2O_5}$ was $1.8\pm1.1\times10^{-4}\ s^{-1}$.
The ACSM submicron aerosol composition data are shown as a time series in Figure 5B and as
a function of time of day in Figure 6. Consistent with the size distributions, mass loadings were
also modest overall (average 2.3 $\mu g\ m^{-3}$). The ACSM factor analysis identified oxygenated



organic aerosol (OOA) as the largest mass fraction of the non-refractory aerosol (average ±
standard deviation 1.4±1.2 µg m$^{-3}$, 63.3% of the total aerosol mass). Hydrocarbon-like organic
aerosol (HOA) associated with primary emissions was a minor component (average
0.03 µg m$^{-3}$, 1.1%) but occasionally enhanced in plumes (maximum 8.3 µg m$^{-3}$).
The oxygenated aerosol fraction (OOA) did not exhibit a discernible diurnal profile (Figure
6A), which is consistent with the modest photochemistry at this site as judged from the modest
peak $O_3$ levels observed.
The inorganic mass fraction was dominated by nitrate (0.47±0.40 µg m$^{-3}$, 20.7%). The second
most abundant inorganic component was ammonium (0.2±1.4 µg m$^{-3}$, (8.8%) followed by
sulfate (0.15±0.15 µg m$^{-3}$, 6.8%). The data are of similar magnitude as aerosol mass
spectrometry (AMS) data collected at nearby Langley as part of Pacific 2001 (Boudries et al.,
2004); then, organics had also been the largest component (average of 1.6 µg m$^{-3}$, 49%), though
sulfate and ammonium mass loadings had been larger (0.88 and 0.44 µg m$^{-3}$, 25% and 14%,
respectively) and nitrate mass loadings smaller (0.38 µg m$^{-3}$, 12%).
The aerosol was frequently neutralized; the neutralization ratio, NR=[$NH_4^+$]:([$NO_3^-$]+[$SO_4^{2-}$])
was 1.38 (median value). The high $NH_3$ content is qualitatively consistent with the non-
quantitative data collected by Metro Vancouver (using a Thermo Scientific 17i
$NH_3$/NO/$NO_2$/$NO_x$ analyzer), which showed large concentrations of gas-phase $NH_3$ (data not
shown).
The ACSM software also identified chloride with an average (±1 standard deviation)
concentration of 0.01±0.03 µg m$^{-3}$, though it is unclear if this signal was real as it did not vary
over the course of the campaign and was below the stated ACSM detection of limit of 0.2 µg m$^{-}$
$^3$ (Ng et al., 2011).
Aerosol nitrate exhibited a clear diurnal profile with higher concentrations at night (Figure 6B).
In particular, the amount of aerosol nitrate increased at the beginning of the night, when the
nocturnal $NO_3$ production rates were greatest.
Previous AMS measurements in Vancouver as part of Pacific 2001 reported a slightly higher
total mass loadings of 7.0 µg m$^{-3}$ that included a greater HOA component (2.4 µg m$^{-3}$, 34%)
and a smaller nitrate fraction (0.6 µg m$^{-3}$, 8.5%) (Alfarra et al., 2004; Jimenez et al., 2009) than
observed here. The lower HOA in this data set are likely a result of tighter emission controls
implemented since the earlier study, a topic outside the scope of this paper.




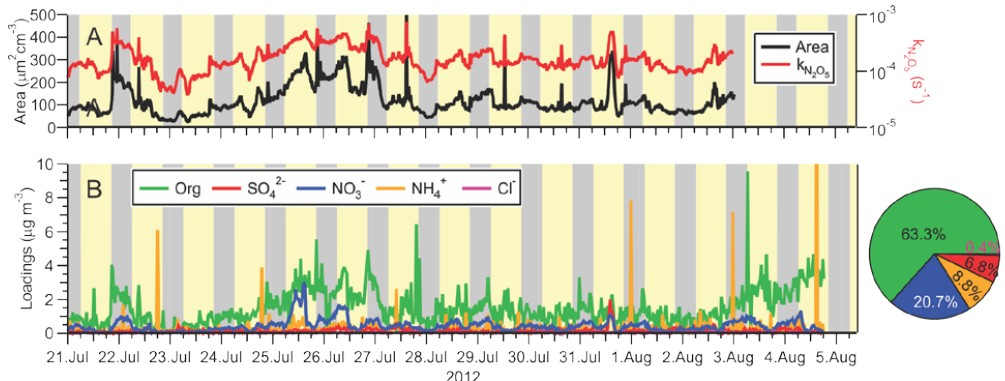


**Figure 5**. Time series of (A) submicron surface area density measured by the TSI 3034 scanning

mobility particle sizer (lhs) and calculate heterogeneous $N_2O_5$ uptake rate coefficient assuming

$\gamma=0.025$ (rhs), and (B) non-refractory submicron aerosol species measured by ACSM. The

average total loading was 2.3 μg m$^{-3}$. The pie chart shows the average campaign composition.





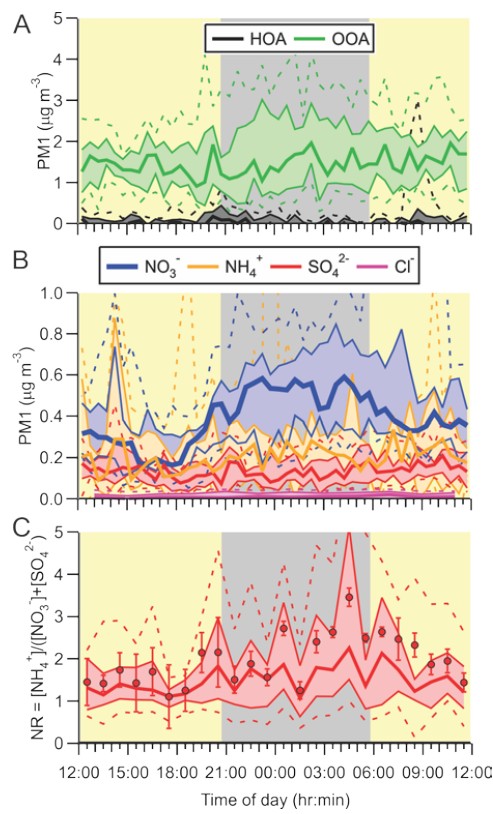


**Figure 6**. Diurnal averages of submicron (PM1) ACMS data. (**A**). Organic aerosol displayed

as hydrocarbon-like organic aerosol (HOA) and oxygenated organic aerosol (OOA) factors. (**B**)

Inorganic aerosol fractions. (C) Neutralization ratio (NR).



### 3.1.6 Hydrocarbon measurements


Mixing ratios of hydrocarbons were quantified during daytime and during the nights of August
2-3 and 3-4. A portion of the hydrocarbon data is shown in Figure 7A.
Mixing ratios were generally smaller during the day than during night, due to the larger daytime
mixing heights. On the nights of August 2/3 and 3/4, $N_2O_5$ was not detected, consistent with
low $P(NO_3)$ values as $O_3$ mixing ratios approached zero (Figure 3). At the same time, there
were strong $NO_3$ sinks present: Mixing ratios of α-pinene and limonene (left-hand axis)
increased throughout the night, as thermal emissions continued into the shallow NBL. In
contrast, mixing ratios of isoprene, whose emissions are driven by photosynthesis (Hewitt et
al., 2011; Guenther et al., 1995), increased at the beginning of the nights and then decreased as
isoprene was removed by oxidation with $O_3$ and $NO_3$ and by transport. Throughout both nights,
the site was also influenced by anthropogenic hydrocarbons (e.g., isooctane and toluene, right-
hand axis). Because synoptic conditions as judged from local wind speed and direction (Figure
3D) were similar on most of the other nights when hydrocarbons were not quantified, the data
shown in Figure 7A were likely representative for much of the campaign.
The VOC data were not sufficiently comprehensive to allow an accurate determination of the
$NO_3$ loss frequency to hydrocarbons, given by $\Sigma k_{NO3+VOC,i}[VOC]_i$. Shown in Figure 7B is the
loss frequency of $NO_3$ to isoprene, calculated by multiplying its concentration with the $NO_3$
rate coefficient taken from *Seinfeld and Pandis* (2006). Loss of $NO_3$ to isoprene was a small
sink compared to its loss to NO via reaction (3) and $NO_3$ photolysis but is approximately on par
with $k_{N_2O_5} K_2[NO_2]$.




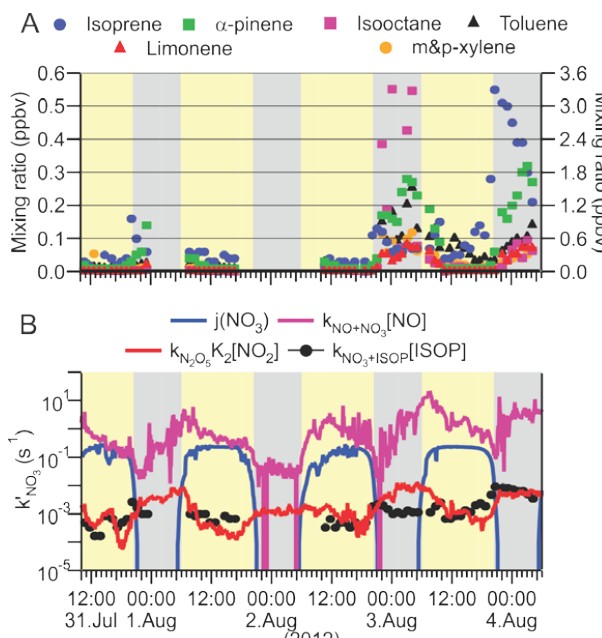


**Figure 7**. (A) Time series of selected VOC mixing ratios observed on the nights of August
2/3 and August 3/4, 2012. Biogenic VOCs (isoprene, α-pinene and limonene) are shown on
the left-hand axis, and anthropogenic VOCs (isooctane, toluene and m&p-xylene) on the
right-hand axis. The α-pinene and limonene measurements are semiquantitative. (B) Time
series of $NO_3$ loss rate coefficients. ISOP = isoprene.



### 3.2 Analysis

#### 3.2.1 Box model simulations of the nocturnal $O_3$ and $O_x$ loss in the NBL

In initial simulations, the $O_3$ and $NO_2$ deposition rates were tuned until the median nocturnal $O_x$ loss was reproduced. An $O_3$ dry deposition rate of $4 \times 10^{-5}$ s$^{-1}$ produced a simulation that reasonably matched the observations (Figure S-1). The magnitude of this rate corresponds to a NBL height of 50 m, the same mixing height that was frequently observed in balloon vertical profiles reported by Pisano et al. (1997). Modeling studies have assumed $N_2O_5$ and $NO_3$ deposition velocities of up to 2 cm s$^{-1}$ in urban areas (Sander and Crutzen, 1996); adopting this value allows the dry deposition rate constants of $N_2O_5$ and $NO_3$ to be estimated at ~$4 \times 10^{-4}$ s$^{-1}$, which is on par with the estimated heterogeneous uptake rate constant of $N_2O_5$ on submicron aerosol.

Next, the generic biogenic VOC was added. For this, a biogenic hydrocarbon abundance of 1 ppbv at sunset (mostly isoprene – see Figure 7) and a (monoterpene) emission rate of $3 \times 10^5$ molecules cm$^{-3}$ s$^{-1}$ based on the crop emission factor given by Guenther et al. (2012) into a 50 m deep NBL were assumed. This assumed flux gives a similar emission rate as the 0.3 ppbv increase over a 6 hour period observed on Aug 3-4 (Figure 7).

The addition of this biogenic VOC only had a marginal effect on $O_x$ and resulted in a slightly better reproduction of the faster $O_x$ loss at the beginning of the night (not shown).

The simulations presented in Figures S-1 underpredict the observed loss of $O_3$, necessitating the addition of an NO source that results in selective removal of $O_3$ while preserving $O_x$. Since automobiles are the largest $NO_x$ source in the region, a constant emission source of 95% NO and 5% $NO_2$ (Wild et al., 2017) was added and its magnitude varied. The $NO_x$ source strength necessary to reproduce the median $O_3$ loss was ~1.1 ppbv hr$^{-1}$. The simulation results using these parameters are superimposed (in red) in Figure 4C. There is reasonable agreement between the simulation and observations until ~3:00, which shows that the nocturnal $O_3$ and $O_x$ loss can be rationalized without active $NO_3$ and $N_2O_5$ chemistry and suggests that $NO_3$, $N_2O_5$, and $ClNO_2$ did not contribute significantly to $O_x$ and $O_3$ loss in the NBL.





### 3.2.2 Metrics of nocturnal nitrogen oxide chemistry: $P(NO_3)$, $\phi'(ClNO_2)$ and $\tau(N_2O_5)$

Nocturnal $N_2O_5$ chemistry was analyzed using several common metrics: the rate of $NO_3$ production by reaction (1), $P(NO_3)=k_1[NO_2][O_3]$, the yield of $ClNO_2$ relative to the total amount of $NO_3$ formed at night, $\phi'(ClNO_2)$, and the steady state lifetime of $N_2O_5$, $\tau(N_2O_5)$.

The time of day dependence of $P(NO_3)$ is shown in Figure 8A. The $NO_3$ production rates were small (median values < 0.3 ppbv hr$^{-1}$) and were larger during the day than at night due to the low $O_3$ mixing ratios. After midnight, for example, the median $P(NO_3)$ was (55±23) pptv hr$^{-1}$. These are very modest $NO_3$ production rates for a site influenced by urban emissions. In a recent study on a mountain top in Hong Kong, for instance, $P(NO_3)$ in excess of 1 ppbv hr$^{-1}$ was observed in polluted air (Brown et al., 2016).

The median integrated nocturnal $NO_3$ production over the course of the night was 940 pptv (Figure 8A, right hand axis), of which 600 pptv were produced before midnight.

The amount of $ClNO_2$ produced relative to this amount, $\phi'(ClNO_2)$, was very small (median 0.17%, maximum 5.4% on the morning of August 2) and considerably less than reported by our group for Calgary (median 1.0%) (Mielke et al., 2016) and Pasadena, CA (median 12%) (Mielke et al., 2013).

A frequently calculated metric of nighttime nitrogen oxide chemistry are the steady state lifetimes of $NO_3$ and $N_2O_5$, $\tau(NO_3)$ and $\tau(N_2O_5)$ (Aldener et al., 2006; Heintz et al., 1996). The latter is calculated from (Brown et al., 2003; Brown and Stutz, 2012):

$$\tau(N_2O_5) = \frac{[N_2O_5]}{P(NO_3)} = \frac{[N_2O_5]}{k_1[NO_2][O_3]} \approx \left( k_{N_2O_5} + \frac{k_{NO_3}}{K_2[NO_2]} \right)^{-1} \tag{11}$$

Here, $k_{N_2O_5}$ and $k_{NO_3}$ are the pseudo-first order loss-rate coefficients of $N_2O_5$ and $NO_3$ respectively, and $K_2$ is the equilibrium constant for equilibrium (2).

The derivation of equation (11) is given by *Brown et al.* (2003). A central assumption in this derivation is that $NO_3$, $NO_2$, and $N_2O_5$ more rapidly equilibrate than $NO_3$ is formed and either $NO_3$ or $N_2O_5$ are destroyed, i.e., $NO_3+N_2O_5$ are assumed to be in steady state with respect to production and loss. *Brown et al.* (2003) outlined potential pitfalls concerning the validity of the steady state approximation and recommended that box model simulations are carried out to



evaluate if a steady state in $N_2O_5$ can be assumed. Using the median nocturnal $NO_2$ and $O_3$
mixing ratios of 7.5 ppbv and 18 to 5.0 ppbv, respectively, a temperature of 286 K, and assumed
$N_2O_5$ and $NO_3$ pseudo-first order loss frequencies of $1\times10^{-3}$ s$^{-1}$ and between $1\times10^{-2}$ s$^{-1}$ and 0 s$^{-1}$,
the time to achieve steady state in $N_2O_5$ is 70 min or less (data not shown). Thus, the steady
state assumption is reasonable for this data set.
A key parameter in equation 11 is the strongly temperature dependent equilibrium constant $K_2$
(Osthoff et al., 2007). At night, the air temperatures during this study were quite warm (median
nocturnal minimum of +13 °C) and did not vary a lot between nights (Figure 8B). The warm
temperatures shift equilibrium 2 away from $N_2O_5$ and towards $NO_3$ and $NO_2$, making losses via
$NO_3$ (reactions 3-4 and 7) more competitive with the losses of $N_2O_5$ (that produce $ClNO_2$), i.e.,
the $\dfrac{k_{NO_3}}{K_2[NO_2]}$ term in equation 11 becomes large relative to $k_{N_2O_5}$. On the other hand, the
relatively high $NO_2$ mixing ratios (median value 7.5±0.8 ppbv) shift the equilibrium towards
$N_2O_5$. Thus, in spite of the relatively warm temperatures, the $N_2O_5$:$NO_3$ equilibrium ratios were
large on aggregate (>15; Figure 8B), enabling $ClNO_2$ formation via reaction 7.
The steady state lifetime of $N_2O_5$, $\tau(N_2O_5)$, is shown as a diurnal average in Figure 8C. The
median $\tau(N_2O_5)$ at night was short (~1 min), and the 90$^{th}$ percentile peaked at a modest 7.6 min
at sunrise, considerably shorter than observed above the NBL (Brown et al., 2006b) and at other
ground sites (Wood et al., 2005; Crowley et al., 2010; Brown et al., 2016)
Superimposed on the right-hand side of Figure 8C are upper limits to the steady state lifetime
of $N_2O_5$ calculated using the titration of $NO_3$ by NO (reaction 3), $NO_3$ photolysis (reaction 4),
and $N_2O_5$ heterogeneous uptake calculated using equation 10, all divided by the $N_2O_5$ over $NO_3$
ratio at equilibrium, given by $K_2NO_2$ (Figure 8B).
$$\tau(N_2O_5) = \left( \frac{k_{NO_3}}{K_2[NO_2]} + k_{N_2O_5} \right)^{-1} < \left( \frac{k_3[NO] + j(NO_3)}{K_2[NO_2]} + k_{N_2O_5} \right)^{-1} \qquad (12)$$
Missing from equation (12) are losses of $NO_3$ to hydrocarbons (which was omitted because of
the poor VOC data coverage) and terms for $NO_3$ and $N_2O_5$ dry and wet (i.e., on cloud and rain
droplets) deposition.



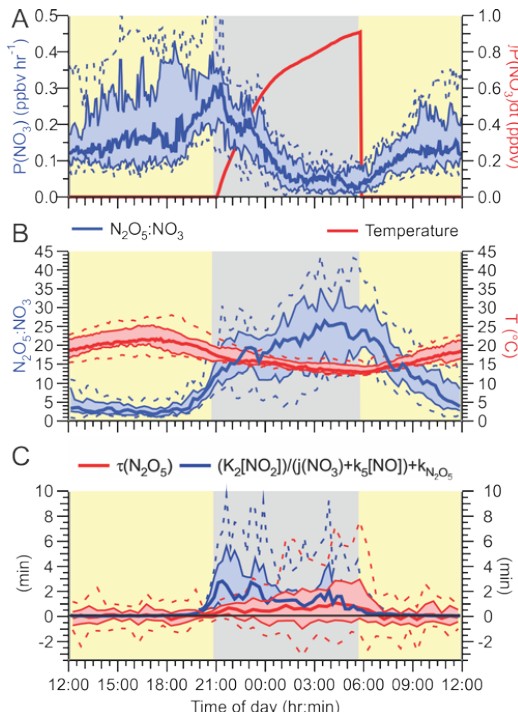


**Figure 8**. (**A**) $NO_3$ production rate $P(NO_3) = k_1[NO_2][O_3]$ as a function of time of day. The red

line is the total amount $NO_3$ generated since sunset, $\int P(NO_3)dt$. (**B**) Equilibrium ratio of $N_2O_5$

to $NO_3$ calculated by multiplying the temperature-dependent equilibrium constant, $K_2$, with the

$NO_2$ concentration, $[NO_2]$ (left axis), and air temperature (right axis). (**C**) Steady state lifetime

of $N_2O_5$ (left axis) and upper limits calculated using equation (12) (right axis) as functions of

time of day.

703

The median "observed" $\tau(N_2O_5)$ is below or equal to the upper limit calculation with equation

12 during both night and day. The largest discrepancy is observed at the beginning of the night,

when oxidation of (unsaturated) hydrocarbons by $NO_3$ (reaction 6) was likely most significant

due to the presence of isoprene. It is also the time when the steady state approximation is most

likely invalid.






### 3.2.3 Heterogeneous conversion of $N_2O_5$ to $ClNO_2$ on the night of August 1/2


Phillips et al. (2016) recently applied several methods to estimate the $N_2O_5$ uptake parameter
($\gamma$) and yield of $ClNO_2$ ($\phi$) from ambient measurements of $NO_3$, $N_2O_5$, $ClNO_2$, and aerosol
nitrate. One of these methods uses the covariance of $ClNO_2$ and aerosol nitrate production rates,
$P(NO_3^-)$ and $P(ClNO_2)$:
$$\phi = 2(P(NO_3^-)/P(ClNO_2) +1)^{-1} \qquad (15)$$
$$\gamma = 2(P(NO_3^-)+P(ClNO_2))/(c\ S_A\ [N_2O_5]) \qquad (16)$$
In the above equation, c is the mean molecular speed of $N_2O_5$ ($\approx 237$ m s$^{-1}$). The use of equations
(15-16) assumes that the relevant properties of the air mass are conserved (i.e., identical upwind
of and at the measurement location and affected identically by air masses mixing), that losses
of measured species are not significant, that the efficiency of $N_2O_5$ uptake and production of
$ClNO_2$ and $NO_3^-$ is independent of particle size, and the absence of partitioning of $HNO_{3(g)}$ and
aerosol nitrate between the gas and particle phases (Phillips et al., 2016). It is assumed further
that production of nitrate on refractory aerosol (that the ACMS does not quantify) is minimal.
In this data set, $ClNO_2$ and submicron aerosol nitrate rarely covaried (Figure 9); the only
instance showing a modest correlation (r=0.66) is the time period prior to sunrise of August 2
(shown as red dots in Figure 9).




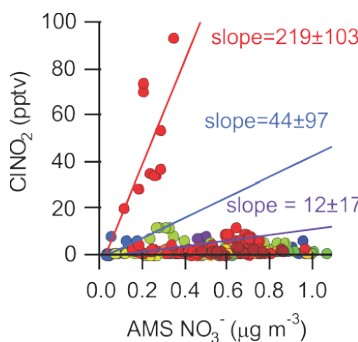


**Figure 9**. Scatter plot of $ClNO_2$ mixing ratios with submicron (PM1) ACMS $NO_3^-$ data. The
slopes were calculated for three periods: Aug 2, 01:25 – 04:55 (red dots; slope = 219±103; $\phi$ =
0.72), July 23, 03:00 – 04:25 (blue dots slope = 44±97; $\phi$ = 0.21), and July 21, 02:25 – 05:20
(purple dots slope = 12±17; $\phi$ = 0.06).

734

The night of August 1-2 exhibited the highest nocturnal nitrogen oxide concentrations for the
entire campaign. Winds were initially from the NW and relatively light (4.8±0.7 km hr$^{-1}$) and
after 01:00 picked up in speed (to 8.2±1.3 km hr$^{-1}$) and shifted to the W. Judging from the
HYbrid Single-Particle Lagrangian Integrated Trajectory (HYSPLIT) back trajectories (Draxler
and Rolph, 2013), the upwind air had moved in from the coast, roughly from the direction of
the city of Victoria, BC (Odame-Ankrah, 2015).

After sunset at ~21:00 local time, $N_2O_5$ levels started increasing and continued to increase until
about 01:30 (Figure 3A). The steady state $N_2O_5$ lifetime at this time was the highest of the
campaign, ~10 min. At 01:20, $ClNO_2$ mixing ratio increased from 20.4 pptv at 01:25 to
93.7 pptv at 04:55 and the aerosol nitrate content from 0.10 to 0.34 μg m$^{-3}$ (40 to 127 pptv).
During this time, $N_2O_5$ mixing ratios and $PM_1$ surface area density were relatively constant,
11.1±6.4 pptv and 67±4 μg m$^{-3}$ (average ± standard deviation), respectively. The combined
amount of $N_2O_5$, $ClNO_2$ and $NO_3^-$ produced (172 pptv) is less than the amount of $NO_3$ produced
from reaction (1) which was 519 pptv during this period.

From equations (15) and (16), a $ClNO_2$ yield of $\phi$ = 0.72±0.34 and an $N_2O_5$ uptake probability
of $\gamma$ = 0.15±0.07 were calculated for this period. Both of these values are upper limits because



production of ClNO$_2$ from uptake of N$_2$O$_5$ on unquantified supermicron (i.e., > 0.5 μm) or
refractory aerosol (which takes place simultaneously) is not accounted for.
A γ value of > 0.05 is greater than can be rationalized from laboratory and field studies (Chang
et al., 2011) and is hence unrealistic. This suggests that ClNO$_2$ production took place
predominantly on supermicron or refractory aerosol, which likely was comprised of mainly sea
salt derived aerosol on this night.
On the other hand, if one assumes that all of the ClNO$_2$ is produced on supermicron or refractory
aerosol such that P(ClNO$_2$) on submicron aerosol equals 0 pptv s$^{-1}$ (which is not unreasonable
considering the absence of measurable amounts of aerosol chloride in this size fraction, see
section 3.1.5), a γ value of 0.08±0.04 is calculated. This large value suggests very efficient
N$_2$O$_5$ uptake (and conversion to aerosol nitrate) on the non-refractory submicron aerosol that
night.

**3.3   Impacts of ClNO$_2$ on radical production**
Photolysis of ClNO$_2$ increases the rates of photochemical O$_3$ production (and hence worsen air
quality) by producing NO$_2$ and reactive Cl atoms (reaction 6). The amounts of ClNO$_2$ available
for photolysis in the morning (median 3.5 pptv at sunrise and 6.8 pptv at 08:00 local time) were
too small to have had a measurable impact on local NO$_2$ concentrations (Figure 3C) but were
sufficiently large to, at least occasionally, impact radical budgets.
Figure 10 shows the instantaneous radical production rates of Cl and OH,
P(Cl)=j(ClNO$_2$)×[ClNO$_2$] and P(OH) from reaction of O($^1$D)+H$_2$O. The latter was calculated
from an assumed steady state in O($^1$D) with respect to its production from O$_3$ photolysis and
reactions with N$_2$, O$_2$, and H$_2$O as described by *Mielke et al.* (2016). This analysis does not
account for OH radical production from photolysis of nitrous acid or aldehydes and, hence,
overestimates the importance of Cl radicals.





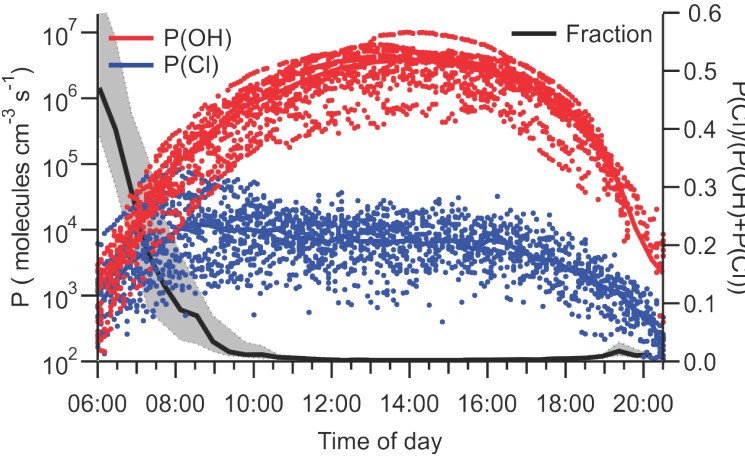

**Figure 10**. Plots of instantaneous rates of Cl (blue) and OH (red) radical production from $ClNO_2$ photolysis and reaction of $O^1D$, generated from $O_3$ photolysis, with $H_2O$ and as a function of time of day. The fraction of radicals produced from $ClNO_2$ photolysis is shown in black. The solid line indicates median values, and shaded areas the 75th and 25th percentiles.

The largest P(Cl) values were observed on July 26, 07:45 local time ($9.5\times10^4$ atoms $cm^{-3}$ $s^{-1}$), accounting for 40% of the total radical production. The largest fraction of radicals produced from $ClNO_2$ photolysis was observed on the same day at 6:35 local time (74%, $7.8\times10^3$ atoms $cm^{-3}$ $s^{-1}$). The photolysis of $ClNO_2$ produces a median value of $6.5\times10^3$ atoms $cm^{-3}$ $s^{-1}$ during daytime, which is negligibly small compared to the median P(OH) of $3.8\times10^6$ molecules $cm^{-3}$ $s^{-1}$ at noon.




## 4    Discussion


It is now well-established that $ClNO_2$ is an abundant nitrogen oxide in many regions of the
troposphere (Table 3). The results presented in this paper are atypical in that they show
consistently small $ClNO_2$ mixing ratios in spite of close proximity to sources, i.e., in a region
where nearby oceanic emissions of sea salt aerosol and $NO_x$ emissions from a megacity
combine. In the following, factors contributing to the low $ClNO_2$ mixing ratios observed in this
study and broader implications of $ClNO_2$ in the LFV are discussed.
The main reason for the low $ClNO_2$ mixing ratios observed in this work are the low nocturnal
mixing ratios of $O_3$ and small $NO_3$ production rate, $P(NO_3)$, resulting from the stratification of
the boundary layer at night and decoupling of the shallow NBL from the NRL. In the following,
it is assumed that a boundary layer structure similar to those observed during PACIFIC 93
(Pisano et al., 1997; McKendry et al., 1997; Hayden et al., 1997) also existed on most
measurement nights of this study. Once the nocturnal boundary layer formed at sunset, $O_3$ and
$O_x$ in the NBL were rapidly (lifetime of $\sim$ 4 hours) removed. The box model simulations
presented in section 3.2.1 show that this removal can be rationalized by dry deposition and
titration of $O_3$ with NO and biogenic VOCs alone, leaving little room for nitrogen oxide
chemistry to destroy $O_3$ or $NO_2$, for example, via heterogeneous formation of HONO which
destroys $NO_2$ (Bröske et al., 2003; Stutz et al., 2004a; Indarto, 2012) or formation of $N_2O_5$ and
subsequent heterogeneous hydrolysis which consumes 2 molecules of $NO_2$ and 1 molecule of
$O_3$ (Brown et al., 2006a). It is the often complete absence of $O_3$ at night which distinguishes
this data set from the other measurement locations for which $ClNO_2$ data have been reported,
including continental sites where aerosol chloride is likely less abundant (Table 3).
A compounding factor in this study was the occasional formation of fog and occasional
precipitation events. Fog droplets act as a very rapid sink for $NO_3$ and $N_2O_5$ (Osthoff et al.,
2006), which shuts down $ClNO_2$ production, and may have also directly contributed
episodically to $ClNO_2$ losses, for example on the morning of July 27. Overall, though, the
contribution of fog to $ClNO_2$ losses in this data set was minor, as only 5% of the measurement
period was impacted by fog. However, this potential $ClNO_2$ loss mechanism should be
investigated further in future lab studies.
The rapid drop of $ClNO_2$ mixing ratio at around 06:00 of Aug 2 is interesting in that it coincided
with a very brief precipitation event. Though an air mass shift cannot be ruled out, this
coincidence suggests the possibility that scavenging of $ClNO_2$ by rain droplets followed by





hydrolysis may be a possible loss pathway. Scavenging of $NO_3$, $N_2O_5$, and $ClNO_2$ by rain
droplets is currently not constrained by laboratory investigations (unlike other gases, such as
$SO_2$ or $NH_3$ (Hannemann et al., 1995)). Similarly to fog, precipitation was not a major factor in
this data set as it affected only 10% but may be in other locations or seasons that experience
higher rainfall amounts.
An important observation is the lack of non-refractory submicron aerosol chloride (Figure 5B).
This suggests that there was limited redistribution of chloride from acidification of sea salt
aerosol onto other aerosol surfaces in this data set. Such a redistribution was observed, for
example, during the Calnex-LA campaign (Mielke et al., 2013). This in turn implies that the
submicron aerosol surface did not significantly participate in the production of $ClNO_2$ from
$N_2O_5$ uptake in the NBL, broadly consistent with the conclusions in section 3.2.3.
The low observed $\tau(N_2O_5)$ levels are consistent with earlier studies that reported strong vertical
gradients in $\tau(N_2O_5)$ due to elevated near-surface sinks from emissions by plants (i.e.,
monoterpenes) and automobiles (i.e., NO and butadiene (Curren et al., 2006)) that titrate $NO_3$
(Stutz et al., 2004b; Wang et al., 2006; Brown et al., 2007; Young et al., 2012). An emblematic
example is the study by *Wood et al.* at a ground site east of the San Francisco Bay Area in
January 2004: They observed relatively modest $N_2O_5$ mixing ratios of up to 200 pptv,
corresponding to $\tau(N_2O_5) < 5$ min for the entire study period (Wood et al., 2005). Studies for
which vertically resolved data were available (e.g., (Stutz et al., 2004b; Wang et al., 2006;
Brown et al., 2007; Young et al., 2012; Tsai et al., 2014) generally showed higher $N_2O_5$
concentrations and hence larger $\tau(N_2O_5)$ aloft in the NRL than at the surface.
A different scenario likely played out aloft in the NRL, which would exhibit higher $NO_3$
production rates (via reactions 1) than the surface layer. Assuming levels of 20 ppbv of $O_3$ and
$NO_2$ in the NRL (Pisano et al., 1997; McKendry et al., 1997), the $NO_3$ production rate would
equal ~1.1 ppbv hr$^{-1}$ in the NRL, roughly on par with values recently reported for Hong Kong,
the current record holder for $ClNO_2$ mixing ratios (Brown et al., 2016; Wang et al., 2016).
Recent aircraft and tower studies have shown high rates of production of $ClNO_2$ aloft (Riedel
et al., 2013; Young et al., 2012), which likely also occurred in this work.
In contrast, the low mixing height of the NBL is conducive to high levels of biogenic
hydrocarbons (section 3.1.6). The nocturnal temperatures during this study were quite warm
and did not vary a lot between nights (Figure 8B). Emissions of monoterpenes, which are
reactive towards $NO_3$, are driven by a temperature-dependent process from storage tissue within



the plants at night (Guenther et al., 1995) and, hence, were likely substantial. Their presence is
likely responsible for some of the gap between the low "observed" $N_2O_5$ steady lifetimes,
$\tau(N_2O_5)$, compared to the upper limit set by reactions 3-4. Even if one assumes a relatively large
uptake probability of $\gamma=0.025$ and accounts for the large ratios of $N_2O_5:NO_3$, the loss rate of
$N_2O_5$ on submicron aerosol was likely small in comparison to losses via $NO_3$ for most of this
data set (Figure 7B). Hence, only a small fraction of the integrated nocturnal $NO_3$ production
of 940 pptv resulted in $ClNO_2$ formation at the surface.
Because of the relatively long lifetime of $ClNO_2$, the breakdown of the surface layer and
merging of the surface air with the NRL constituted itself as a $ClNO_2$ "morning peak" in a
similar manner as what has recently been reported at other locations (Tham et al., 2016; Bannan
et al., 2015; Faxon et al., 2015). This morning peak is rationalized by higher net $ClNO_2$
production in the NRL; the break-up of this layer ~2 hours after sunrise then mixes $ClNO_2$ down
to the surface. Such a vertical mixing process was not seen during Calnex-LA (Young et al.,
2012; Tsai et al., 2014) where the NBL was sufficiently deep to prevent complete $O_3$ removal
and the $ClNO_2$ produced mixed down to the surface at night.
Assuming a 100 m deep NRL where $ClNO_2$ production takes place, a mixed layer height of
500 m by 08:00 (Pisano et al., 1997) and negligible destruction of $ClNO_2$ by photolysis (which
is reasonable as the lifetime of $ClNO_2$ with respect to photolysis is >4.6 hours at that time of
day), a morning increase in $ClNO_2$ mixing ratio by 40 pptv at the surface as seen on the morning
of July 26 suggests a pool of $ClNO_2$ in the NRL at sunrise of ~200 pptv, likely a modest value
considering that the (assumed) $NO_3$ production rate may have integrated to ~9 ppbv over the
course of the night.
The largest nocturnal $ClNO_2$ mixing ratios were observed on July 22/23 and August 1/2. Both
of these nights exhibited high wind speeds and are counterexamples to what was observed on
other nights. We speculate that the higher levels of wind shear and turbulence altered the
nocturnal boundary layer structure which exhibited a greater degree of vertical mixing and
higher $O_3$ concentrations at the surface. Consistent with this interpretation and the notion that
an isolated NRL with higher net $ClNO_2$ production was absent on those nights, the mornings of
July 23 and Aug 2 did not show a "morning peak". In contrast, low surface wind speeds were
observed on the other nights, facilitating a stable and shallow nocturnal surface layer.
It is conceivable that a land-sea breeze effect transported air from a region closer to the coast
that saw higher $ClNO_2$ production than at Abbotsford, i.e., that the $ClNO_2$ morning peaks are




generated by horizontal as opposed to vertical transport. Large $NO_3$ mixing ratios have been
reported at Saturna Island, which strongly suggest that sizeable reservoirs of $ClNO_2$ form
offshore at night. However, it is known how far inland these reservoirs extend. Considering the
average wind speed in the morning (6 km $hr^{-1}$), distance to the coast (35 km), and close
proximity (200 m) of the site to the bottom of the polluted NRL with documented high nocturnal
pollution levels and early morning down mixing events, the vertical transport explanation is
much more likely correct.  Nevertheless, measurements of $ClNO_2$ at a site closer to the coast
(e.g., at White Rock) would be beneficial.
Formation of $ClNO_2$ affects air quality through its photolysis which generates $O_x$, $NO_x$, and
reactive Cl radicals in the morning, leading to higher net photochemical $O_3$ production (Sarwar
et al., 2014). In spite of the low levels of $ClNO_2$ observed in this work, the production of radicals
from its photodissociation was not always negligible (Figure 10). Conditions leading to $O_3$
exceedances did not develop during this study. If such conditions had developed, it is highly
likely that this radical generation would have played a much greater role.
The data presented here suggest that higher rates of $ClNO_2$ and subsequent radical generation
take place routinely in layers aloft, processes that are not directly observable at the surface but
whose implications are felt as the ultimate product, $O_3$, is sufficiently long-lived to mix down
to the surface (McKendry et al., 1997). Future studies should therefore target the NRL, for
example through missed-approaches by aircraft, a blimp, or from a tall tower, especially during
episodes of a developing $O_3$ exceedance event.



## 5   Summary and conclusions


In this paper, we have presented the first measurements of $ClNO_2$ and $N_2O_5$ mixing ratios in
the LFV. In spite of the close proximity to $NO_x$ (megacity of Vancouver) and sea salt aerosol
(the Pacific Ocean) sources, $ClNO_2$ and $N_2O_5$ mixing ratios were small (maximum of 97 and
27 pptv, respectively) and smaller than observed at other measurement locations for which
$ClNO_2$ abundances were reported. The low mixing ratios are explained through the removal of
$O_3$ by deposition and titration with NO in a shallow nocturnal surface layer. Measurements of
submicron aerosol composition by ACMS showed no enhancements of particle-phase chloride,
which is in contrast to locations where high $ClNO_2$ mixing ratios were observed (such as
Pasadena (Mielke et al., 2013)) and indicates that there was little processing and redistribution
of sea salt derived chloride at this location. There is indirect evidence that higher production of
$ClNO_2$ took place above the measurement site in the NRL, observed via downmixing after the
break-up of the NBL in the morning, and highlights the need for future vertically resolved
measurements (e.g., from an aircraft platform) of $ClNO_2$ and $N_2O_5$ mixing ratios in the LFV.
Conditions leading to $O_3$ exceedences did not develop during the relatively short measurement
period of 2 weeks, such that the full impact that nocturnal formation of $ClNO_2$ could have on
radical production and $NO_2$ recycling remains unquantified.





**Data availability**
The data used in this study are available from the corresponding author upon request
(hosthoff@ucalgary.ca).

**Acknowledgments**
This project was undertaken with the financial support of the Government of Canada through
the Federal Department of the Environment. Ce projet a été réalisé avec l'appui financier du
Gouvernement du Canada agissant par l'entremise du ministère fédéral de l'Environnement.
Partial funding for this work was provided by the Natural Sciences and Engineering Research
Council of Canada (NSERC) in the form of operating ("Discovery") and Research Tools and
Instruments (RTI) grants. The Abbotsford field study was financially supported by a BC Clear
research grant from the Fraser Basin Council of British Columbia and by Metro Vancouver.




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

**Table 1**. Summary of measurement techniques deployed at T45 during the study.

| Species or parameter | Method | Uncertainty | Time resolution |
|---|---|---|---|
| $ClNO_2$, PAN, PPN | Chemical ionization mass spectrometry (Mielke et al., 2011) | ±25% ±10% | 30 s |
| $N_2O_5$ | Red diode laser cavity ring-down spectroscopy (Odame-Ankrah and Osthoff, 2011) | ±25% | 1 s |
| $O_3$ | UV absorption (Thermo 49i) | ±10% | 10 s |
| NO/$NO_y$ | $O_3$-Chemiluminescence (Thermo 42i-Y) with heated Mo converter; operated with inlet filter | ±30% | 10 s |
| $NO_2$ | Blue diode laser cavity ring-down spectroscopy (Paul and Osthoff, 2010) | ±10% | 1 s |
| PAN, PPN | Gas chromatography with electron capture detection (Tokarek et al., 2014) | ±10% | 6 min |





| Photolysis frequencies | Spectral radiometry (Metcon) | ±20% | 10 s |
|---|---|---|---|
| Aerosol size distribution | Scanning mobility particle sizer (SMPS) | | nd |
| Aerosol composition | Aerosol Chemical Speciation Monitor (ACSM) | ±20% | 30 min |
| VOCs | Agilent | ±30% | 20 min (1 hr*) |
| Meteorological data | Various | | |

* Sampled for 20 min within a 1 hour time period



**Table 2**. Ratios of up- to down-dwelling photolysis frequencies.

| Frequency | Ratio |
|:---:|:---|
| $j(NO_3)$ | 0.27±0.04 |
| $j(NO_2)$ | 0.15±0.03 |
| $j(ClNO_2)$ | 0.14±0.02 |
| $j(O_3 \rightarrow O(^1D))$ | 0.11±0.02 |




**Table 3**. Maximum ClNO$_2$ mixing ratios observed to date.

| Location | Type | Maximum mixing ratio | Reference(s) |
|---|---|---|---|
| Houston, TX | Off-shore, costal, and inland | 1.2 ppbv | (Osthoff et al., 2008) |
| New England | Off-shore | 90 pptv | (Kercher et al., 2009) |
| Pasadena, CA | Off-shore | 2.15 ppbv | (Riedel et al., 2012a) |
| La Jolla, CA | Coastal | 30 pptv | (Kim et al., 2014) |
| Boulder, CO | Continental | 425 pptv | (Thornton et al., 2010) |
| Calgary, AB | Continental | 330 pptv | (Mielke et al., 2016; Mielke et al., 2011) |
| Erie, CO | Continental | 1.3 ppbv | (Riedel et al., 2013; Brown et al., 2013) |
| Feldberg, GER | Continental | 800 pptv | (Phillips et al., 2012; Phillips et al., 2016) |
| Horsepool, UT | Continental | 500 pptv | (Edwards et al., 2014) |
| Pasadena, CA | Coastal, inland | 3.5 ppbv | (Mielke et al., 2013) |
| London, UK | Coastal, inland | 724 pptv | (Bannan et al., 2015) |
| Hongkong, PRC | Coastal, inland | 2.0 ppbv | (Tham et al., 2014) |
| Southeast TX | Coastal, inland | 280 pptv | (Faxon et al., 2015) |
| Hongkong, PRC | Coastal, inland | 4.7 ppbv | (Wang et al., 2016) |
| North China Plain | Continental | 2.1 ppbv | (Tham et al., 2016) |
| North China Plain | Continental | 776 pptv | (Wang et al., 2017) |
| Abbotsford, BC | Coastal, inland | 97 pptv | This work |
