# Peer review of "nitrogen oxides in the Lower Fraser Valley of British"

_Atmospheric Chemistry and Physics, 2017_

## Referee Comment (RC1) · Anonymous Referee #2 · 23 Jan 2018

This manuscript reports measurements of ClNO2, N2O5 and other chemicals (ozone, NOx, NOy, aerosol size and composition and VOCs etc) at a surface site near the Lower Fraser Valley during a two-week period in summer 2012. The study was motivated by the need to investigate the role of ClNO2 in ozone exceedance in the region. However, the relatively short field study did not capture any high ozone events, and low ClNO2 levels were observed due to fresh emissions of NO which suppress the production of N2O5 and ClNO2 at nigjt. The paper investigated some metrics related to production and loss of N2O5/ClNO2 with the aid of a simple box model, and the results show small contribution of ClNO2 to radical production in such an environment, as one would expect. While the data on ClNO2 and N2O5 add to the global data base of the

two important and poorly documented species, the main finding (low N2O5/ClNO2 in high NO condition and resulting small contribution of photolysis of ClNO2 to radical source) gives limited new insight on the processes and impact of N2O5 and ClNO2, as such the significance of this work is unclear.

Specific questions on methods: What was the extent of N2O5 loss in the sampling line during the field study? For NOy measurements, was the Mo converter placed at the sample inlet outside? Was a filter placed before the Mo converter? The aerosol size measurements only covered size10 nm to 487 nm, were the larger size particles considered when calculating the aerosol surface areas density? What was the uncertainty of the simple box model adopted?

---

## Referee Comment (RC2) · Anonymous Referee #1 · 12 Feb 2018

Osthoff et al present a thorough study of NOy composition in the Lower Fraser Valley in British Columbia where air quality episodes can occur, but did not during their study. Notably, despite being a coastal site, low levels of ClNO2 were observed due to limited nocturnal NOx chemistry. A comprehensive description of the results is provided. My main comments below surround the discussion of the aerosol data and the presentation/formatting of the main text. A full list of detailed comments is provided below.

Major Comments: There are numerous places in the manuscript where 1-2 sentence "paragraphs" exist (Section 2.2 and elsewhere throughout); these sentences should be

integrated in longer paragraphs for improved flow. Currently, this makes the manuscript difficult to read, and it also makes appears sloppy. I disagree with the authors that these revisions should wait until "the type setting stage" (authors' response to Quick Review), as I believe that it significantly impacts the presentation of the results and discussion. Similarly, please refer to reaction numbers in the text when the reactions are presented (e.g. page 4 and elsewhere).

There are many places in the text that state "(not shown)" with respect to results and ask the reader/reviewer to trust the authors; it would be more helpful for the reader's evaluation of the results for these data to be presented in the supplementary information.

Section 2.5: More information is needed for the description of the box model. A list of reactions should be provided in the supplementary information. Is chlorine chemistry included? Are aerosols included?

Lines 443-445: This sentence can be strengthened by referring at least to the timing of the ozone maximum for support, and perhaps referring to the next section. Otherwise it sounds like a guess that you cannot support further, which is not true.

Section 3.1.5: This section needs the most revision, particularly with respect to the presentation and discussion of the ACSM data. The authors quantify fractions of "total aerosol mass" (e.g. line 541); however, only non-refractory submicron aerosol was measured. It is expected at refractory sea salt aerosol contributes significantly to this site, so these calculations are expected to be inaccurate. Similarly, the authors discuss the "inorganic mass fraction" and "most abundant inorganic component", which also are influenced by refractory aerosol, such that the mass fractions are expected to be inaccurate. The discussion of the ACSM data must clearly reflect that only submicron non-refractory mass was measured and that sea salt aerosol (most relevant for ClNO2 production) was not measured.

Lines 509-517: It should be clarified that this only reflects the aerosol <0.5 um in

diameter, based on the size range measured by the SMPS. Since aerosol surface area peaks at a higher diameter than aerosol number, it would be expected that this calculation of aerosol surface area may be a significant underestimate. This should be stated, and the implications of this should be discussed where appropriate.

Lines 554-558: The authors should refer to Zhang et al (2007, Environ. Sci. Technol.) for the proper method for examining aerosol acidity using ACSM data. Please show the gas-phase NH3 data, at least in the supplementary information.

Lines 559-562 and Figure 6 caption: Only non-refractory chloride was monitored by the ACSM, and this should be noted, given the importance for ClNO2 production. If the signal was below the instrument limit of detection, then the concentration calculated is, by definition, not accurate. The text should be revised to reflect these two important points.

Lines 566-570: The time of year is expected to be quite important for these comparisons. Please provide this information in the discussion.

Lines 616 & 621: Do these dry deposition rates make sense in the context of previously published literature?

Lines 629-630: Why is this data not shown? It is about modeling the "faster Ox loss at the beginning of the night", which seems central to the section header "Box model simulations of the nocturnal O3 and Ox loss in the NBL".

Lines 632-634: Why was the model not simply constrained to measured NO? Did NO under these modeled conditions match what was measured?

Lines 692-694: Not including loss of NO3 to hydrocarbons is not justified here. Why not assume a generic BVOC as you did in an earlier section, or use the data that you do have? These options seem better than blindly ignoring this NO3 loss process. Further, on lines 706-707, you state that reaction between NO3 and isoprene was likely significant, suggesting that that reaction should be included, especially since the

authors have at least some measurements of isoprene. Also, not including NO3 and N2O5 deposition is not justified. It seems that dry deposition could be easily included. Where time periods that fog and rain occurred used? If so, these periods should be removed for these calculations.

Lines 721-723: It is assumed that the production of nitrate on refractory aerosol is minimal, but this is not justified and is a poor assumption. Nitric acid displacement of HCl is one of the most common sea salt aerosol aging pathways (Gard et al. 1998, Science).

Line 757-762: Only non-refractory chloride and nitrate were measured! This should be considered and reflected in this discussion.

Line 827: How does this observation of a lack of non-refractory submicron aerosol chloride compare to other similar inland coastal AMS/ACSM studies?

Lines 913-916: The authors suggest that the lack of particle-phase chloride (should be 'non-refractory' chloride) is in contrast to their previous study, Mielke et al. 2013, in Pasadena, CA. However, in that previous paper, AMS PM1 showed very little non-refractory chloride, with far higher levels of PM2.5 chloride (refractory + non-refractory measured by PILS-IC) measured. So, this is not a complete comparison, and in fact, in terms of non-refractory PM1 chloride, the studies seem fairly similar. This discussion should be reconsidered and revised.

Table 1: Please add the SMPS and ACSM size ranges, as well as note that the ACSM aerosol composition reflects only the "non-refractory" aerosol.

Minor Comments: Line 65: Fix typo – "particle" should be "particulate".

Reaction 6 should include chloride as a reactant.

Please add references to the following lines: 77, 99, 108, 426, 431, 887.

Lines 135-136, 837: Fix reference formatting in sentence.

[Figure]

Line 337: Fix typo – "day used" should likely be "day were used".

Section 3.4, Table 2, & anywhere else: "down-dwelling" and "up-dwelling" should be "down-welling" and "up-welling".

Line 457 & elsewhere: Error should be given as one significant figure, with the average value provided with the same number of decimal places. For example, line 457 should list 64 +/- 1 ppbv, rather than 64.4 +/- 1.2 ppbv.

Lines 459-461: Provide values in parentheses for context.

Line 467: Change ":" to "."

Line 469: Fix typo – "loss of are" should likely be "loss are".

Line 482: Fix typo – "at a median value" should be "at median values".

Lines 483-484: Fix typo – "ratio of this campaign was" should be "ratios of this campaign were".

Figure 5: Please clarify this figure caption. It was not obvious at first what "lhs" and "rhs" stood for, as these acronyms are not defined.

Lines 601-603, 854-859: Please revise sentences to improve clarity.

Line 658: Fix typo - "are" should be "is".

Line 664: Delete sentence as this information is already given on line 660.

Lines 730 & 914: Fix typo – "ACMS" should be "ACSM".
* * *

---

## Referee Comment (RC3) · Anonymous Referee #3 · 15 Feb 2018

General Comments:

This is a well written manuscript describing studies of nocturnal chemistry in the Lower Frasier Valley, a region near a megacity (Vancouver) and with sea salt sources. The combination of these pollution (NOx) sources and sea salt aerosol particles might be expected to produce nitryl chloride. In fact, Pasadena, CA, in the Los Angeles area (also a megacity near the ocean) had much larger nitryl chloride at ground level. The authors argue that shallow boundary layers, titration of ozone by fresh NO emissions at ground level and potentially biogenic VOC inputs often preclude nitryl chloride formation at ground level, which is supported by the data in the manuscript. The work is

well written, sufficiently referenced, and appropriate for publication in ACP.

There is a lot of evidence in this manuscript that titration of ozone at ground level was a reason for low NO3 and N2O5 levels. On the nights when there was ozone, N2O5 and ClNO2 were at their largest mixing ratios. Presumably this titration is caused by input of NO at ground level, which does not mix to very high altitude. Therefore, it is likely that aloft there is more active N2O5 chemistry and probably ClNO2 production. For this reason, I think that the manuscript's title should be modified to include "Low levels of nitryl chloride at ground level...". The peaking of ClNO2 after sunrise and coincident with breakup of the nocturnal boundary layer would seem to indicate that ClNO2 aloft is likely higher.

The manuscript has discussion about chloride measurements based upon the ACSM, which is not good at detecting chloride in the form of NaCl. There should be measurements in the area of PM2.5 chemical composition that could help to better understand the presence of sea salt chloride. The authors should examine available aerosol chloride measurements to expand their analysis and interpretation through consideration of these data.

Specific comments:

Showing population density (in some way) on the Figure 1 map would be nice.

Figure 3 seems to be mentioned before Fig. 2

Line 194: I think the word is "aging"

Line 199: I don't think ECCC is defined?

Line 202: Define THS?

Line 360: Presumably after measuring the upwelling/downwelling actinic ratio, this ratio was used to correct all downwelling actinic flux data to be a total actinic flux. If this was done, it should be noted.

Line 394: I think K2 is first used here but not defined until later. This should probably be done near line 80

Line 533: This equation is applicable when diffusion limitations are not important, which might not be true if supermicron particle surface area is involved.

Line 554: Make clear that an equivalent basis (e.g. 2x the molar concentration of sulfate) is being used in this neutralization ratio equation.

Line 559-562: Does the ACSM detect chloride efficiently? Standard filter samples would show chloride and could be used to verify its presence or absence. Historical data from the area would tell you the ratio of chloride to other inorganic ions (e.g. nitrate and sulfate), so you should be able to tell if the ACSM is not actually detecting chloride efficiently.

Line 751 area: Presumably some pollution monitoring studies looked at PM2.5 via IC and could address presence of at least ∼1 to 2.5 micron particles containing Cl-

---

## Author Comment (AC1) · 27 Mar 2018

We thank the referees for their thorough review of the manuscript. Their comments are reproduced below in **bold italic font**. Our responses are given in regular font. The line numbers in our responses below refer to the version posted on the ACPD web site.

*Anonymous Referee #1

*Osthoff et al present a thorough study of NOy composition in the Lower Fraser Valley in British Columbia where air quality episodes can occur, but did not during their study. Notably, despite being a coastal site, low levels of ClNO2 were observed due to limited nocturnal NOx chemistry. A comprehensive description of the results is provided.*

*My main comments below surround the discussion of the aerosol data and the presentation/formatting of the main text. A full list of detailed comments is provided below.*

We thank the reviewer for the constructive and detailed feedback.

It is clear from some of the line numbers that this particular reviewer referred to that she/he occasionally commented on an earlier version of the manuscript (the one subject to the "quick review" process; e.g., section 3.1.5, which the reviewer refers to lines 509-517 which changed to lines 527-536). In the version posted for discussion on the ACPD web site, we had already incorporated changes in response to the reviewer's feedback during the "quick review" phase. Hence, some of the reviewer's concerns raised below (such as concerns about what the ACSM quantifies) were already addressed.

As to the concerns regarding formatting (style of reaction numbering, paragraph breaks etc.): please see our comments in the next section.

*Major Comments:*

*There are numerous places in the manuscript where 1-2 sentence "paragraphs" exist (Section 2.2 and elsewhere throughout); these sentences should be integrated in longer paragraphs for improved flow. Currently, this makes the manuscript difficult to read, and it also makes appears sloppy.*

*I disagree with the authors that these revisions should wait until "the type setting stage" (authors' response to Quick Review), as I believe that it significantly impacts the presentation of the results and discussion.*

We appreciate the author's opinion, though we do not share this sentiment. There are no are set rules that guide the length of paragraphs in scientific papers. Guidelines on how paragraphs should be constructed vary considerably between disciplines, and writing is rightfully referred to as an "art" (Plaxco, 2010).

In our opinion, a paragraph should discuss a single idea and thus should have a single, unifying theme running through it; as a result, we generally started a new paragraph whenever the theme changed or deviated and couldn't be simply be tied to the original one. The paragraphs in section 2.2, for example, focus on properties common to all techniques (paragraph 1), properties only common to gas-phase instruments operated from the U Calgary trailer (paragraph 2), and properties only common to the

aerosol and VOC measurements made by Metro Vancouver and ECCC (paragraph 3). In our opinion, the paragraph/line breaks improve clarity and were, in fact, carefully and intentionally (not sloppily) constructed, though we admit that some paragraphs ended up on the short side. We have tried to the best of our ability to make improvements, for example by removing line breaks where they were perhaps not needed.

***Similarly, please refer to reaction numbers in the text when the reactions are presented (e.g. page 4 and elsewhere).***

Done.

***There are many places in the text that state "(not shown)" with respect to results and ask the reader/reviewer to trust the authors; it would be more helpful for the reader's evaluation of the results for these data to be presented in the supplementary information.***

There were four such instances. In all cases, the information not shown are on the periphery of the manuscript, and we felt it unnecessary to needlessly bloat the paper. However, we do not mind adding the information requested to the S.I. and have now done so in most cases.

The first of these instances, on line 424, refers to the average of the $NO_x/NO_y$ ratio for entire campaign:

"The average $NO_x/NO_y$ ratio for the entire campaign was 0.89 (data not shown)."

In the paragraph preceding this sentence, it is noted that the time series of $NO_y$ and of NO and $NO_2$ are shown in Figures 3B and 3C, respectively.  We therefore feel it is unnecessary to also show a time series of the ratio, especially since the ratios carry substantial uncertainties (see Table 1 – NO and $NO_y$ are good to ±30%, $NO_2$ to ±10%): In the absence of NO, the uncertainty in the ratio of $NO_x$ to $NO_y$ is ±40% (and higher still in the presence of NO).

In response to the reviewer's comment, we have removed the phrase "(data not shown)" since the data are shown Figures 3B and 3C but have inserted the uncertainty.

"The average $NO_x/NO_y$ ratio for the entire campaign was 0.9±0.4."

The second instance refers to the $NH_3$ data (line 557). As was stated on line 556, we the $NH_3$ data collected by Metro Vancouver were not quality-assured and hence non-quantitative (in part due to inlet memory effects). Regardless, these are now shown as Figure S-1 in the S.I.:

[Figure]

**Figure S-1**. Time series of gas-phase ammonia data reported by Metro Vancouver. Data were not quality-assured and are non-quantitative.

The third instance (line 630) refers to a box model simulation in which a single reaction was added. We have modified this section as follows

"The addition of this biogenic VOC only had a marginal effect on $O_x$  (Figure S-3)."

and have added the following description of this rather simple model to the S.I.

**Box model to rationalize $O_x$ loss by dry deposition**

A box model was set up to simulate the median nocturnal decays of $O_3$ and $O_x$. These simulations are intended as back-of-the-envelope type estimates of major processes only since an accurate description of the nocturnal boundary layer chemistry would require modeling of horizontal and vertical transport, i.e., altitude-resolved information (Geyer and Stutz, 2004). Such information was not available in this work.

The reactions used in this model are summarized in Table S-2. The mechanism consists of $O_3$ and $NO_2$ dry deposition, titration of NO with $O_3$ (R8) and chemical loss of $O_3$ to a generic biogenic hydrocarbon. For dry deposition, the velocities of $v_d(O_3)$ = 0.2 cm s$^{-1}$ and $v_d(NO_2)$ = $\alpha \times v_d(O_3)$ with $\alpha$=0.65 from Lin et al. (2010) were used. The rate constants for reaction with the generic biogenic hydrocarbon was set to that of $\alpha$-pinene with $O_3$ ($5 \times 10^{-11}$ cm$^3$ molec.$^{-1}$ s$^{-1}$ (Seinfeld and Pandis, 2006)).

Model simulations were carried out using a custom differential equation integrator macro in the software package Igor Pro (Wavemetrics) and were initiated with the campaign median $NO_2$ and $O_3$ concentrations observed at sunset.

**Table S-2**. Reactions included in box model to estimate dry deposition velocities

| Reaction | Rate constant |
| --- | --- |
| $O_3 \rightarrow$ products | $k_{dep}(O_3)$ |
| $NO_2 \rightarrow$ products | $k_{dep}(NO_2)$ |
| $O_3 + NO \rightarrow NO_2 + O_2$ | $4.8 \times 10^{-4}$ ppbv$^{-1}$ s$^{-1}$ |
| $O_3 + VOC \rightarrow$ products | 1.25 ppbv$^{-1}$ s$^{-1}$ |

[Figure]

**Figure S-3**. Effect of biogenic VOC emissions on $O_x$. The observed and simulated $O_x$ loss in the NBL at Abbotsford assuming an $O_3$ dry deposition rate of $4\times10^{-5}$ s$^{-1}$ are shown as green and blue traces, respectively. The red trace shows the effect of adding 1 ppbv of reactive biogenic VOC at sunset and continuous biogenic VOC emissions of $3\times10^5$ molecules cm$^{-3}$ s$^{-1}$ throughout the night.

The fourth (line 672) is a box model simulation to estimate the time to achieve steady state. The methodology has been described by Brown et al. in J. Geophys. Res., 108, 4539, doi: 4510.1029/2003JD003407, 2003. We have added the following text to the S.I.:

**Box model to determine the time necessary for $NO_3$ and $N_2O_5$ to achieve a steady state with respect to production and loss**

The validity of the steady state assumption was evaluated in a similar fashion as described by Brown et al. (2003) using a simple box model. Reactions and rate coefficients included in these simulations are listed in Table S-3. Model simulations were carried out using a custom differential equation integrator macro in the software package Igor Pro (Wavemetrics). Rate coefficients were calculated for a temperature of 286 K, which is the median nocturnal temperature of this study (Figure 8B). Simulations were initiated with the median nocturnal $NO_2$ and $O_3$ mixing ratios of 7.5 ppbv ($1.92\times10^{11}$ molecules cm$^{-3}$) and of either 18 ppbv ($4.5\times10^{11}$ molecules cm$^{-3}$) or 5.0 ppbv ($1.3\times10^{11}$ molecules cm$^{-3}$), respectively. The simulations assume pseudo-first order $N_2O_5$ and $NO_3$ loss with frequencies of $1\times10^{-3}$ s$^{-1}$ and between $1\times10^{-2}$ s$^{-1}$ and 0 s$^{-1}$, respectively.

Simulated temporal profiles of $NO_3$ and $N_2O_5$ are show in Figure S-5 (left axis) and those of $O_3$ and $NO_2$ on the right axis. The subpanels A, B, and C are simulations with $k_{NO3}$ = 0 s$^{-1}$, $1\times10^{-3}$ s$^{-1}$ or $1\times10^{-2}$ s$^{-1}$, respectively. In each case, the rate of change of [$N_2O_5$] with respect to time, d[$N_2O_5$]/dt, approaches zero after a period of ~70 min, or less, indicating the time to approach steady state.

The simulations also show that the amount of $O_3$ and $NO_2$ removed through chemical reactions of $NO_3$ and $N_2O_5$ are ~1 ppbv and between ~1.9 and ~1.6 ppbv over a period of 4 hours. These are upper limits as, in this study, much of the $NO_3$ was titrated by NO. In any case, loss of $O_3$ through nocturnal gas-phase

is predicted to be rather small compared to the total $O_3$ loss observed (~26 ppbv over 9 hours, see section 3.1.3 and Figure 4C in the main text).

Brown et al. (2003) show that in these scenarios, $NO_3$, $N_2O_5$, and $NO_2$ remain in equilibrium almost throughout; for completeness, the corresponding plot for these simulations is shown in Figure S-6.

As shown in equation (2) of the manuscript, the steady state lifetime is approximately equal to:

$$\frac{[N_2O_5]}{k_1[NO_2][O_3]} \approx \left(k_{N_2O_5} + \frac{k_{NO_3}}{K_2[NO_2]}\right)^{-1} \tag{2}$$

A comparison of these two expressions is shown in Figure S-7. The time when these two expressions are equal is equal to the time to steady state.

Table S-3. Reactions included in the box model to estimate the time for $NO_3$ and $N_2O_5$ to achieve steady state with respect to their production and loss

| # | Reaction | Rate coefficient |
|---|----------|------------------|
| R1 | $NO_2 + O_3 \rightarrow NO_3 + O_2$ | $2.28 \times 10^{-17}$ cm$^3$ molecule$^{-1}$ s$^{-1}$ |
| R2$_f$ | $NO_3 + NO_2 \rightarrow N_2O_5$ | $1.35 \times 10^{-12}$ cm$^3$ molecule$^{-1}$ s$^{-1}$ |
| R2$_r$ | $N_2O_5 \rightarrow NO_3 + NO_2$ | $0.00923$ s$^{-1}$ |
| (R7) | $NO_3 \rightarrow$ products | $k_x = k_{NO3} = 0$ s$^{-1}$, $1 \times 10^{-3}$ s$^{-1}$ or $1 \times 10^{-2}$ s$^{-1}$ |
| (R5) | $N_2O_5 \rightarrow$ products | $k_y = k_{N2O5} = 1 \times 10^{-3}$ s$^{-1}$ |

[Figure]

**Figure S-5**. Simulated temporal profiles of $NO_3$ and $N_2O_5$ (left axis) and $O_3$ and $NO_2$ (right axis). The subpanels A, B, and C are simulations with $k_{NO3} = 0$ $s^{-1}$, $1\times10^{-3}$ $s^{-1}$ or $1\times10^{-2}$ $s^{-1}$, respectively.

[Figure]

**Figure S-6**. Equilibrium constants for reaction (2) calculated for the three scenarios shown in Figure S-4.

[Figure]

**Figure S-7**. Comparison of $\tau(N_2O_5)$ calculated using equation (2) of the main manuscript. with the dashed lines calculated using equation (11) of Brown et al. (2003).

**Section 2.5: More information is needed for the description of the box model. A list of reactions should be provided in the supplementary information. Is chlorine chemistry included? Are aerosols included?**

We have added a list of reactions to the S.I. as requested (see above).

It is stated in section 2.5 that "These simulations are intended as back-of-the-envelope type estimates of major processes only since an accurate description of the nocturnal boundary layer chemistry would require modeling of horizontal and vertical transport, i.e., altitude-resolved information not available in this study (Geyer and Stutz, 2004)." Because of the limited scope of these simulations, chlorine and aerosol chemistry was neglected; more importantly, their impact in this study on $O_3$ and $O_x$ at night was likely very minor. No further changes were made to the manuscript.

**Lines 443-445: This sentence can be strengthened by referring at least to the timing of the ozone maximum for support, and perhaps referring to the next section. Otherwise it sounds like a guess that you cannot support further, which is not true.**

We are assuming that the reviewer is referring to the following sentence: "This can be rationalized by a greater abundance of oxidants that oxidize NO to $NO_2$, i.e., $O_3$ (see Figures 3 and 4) and organic peroxy radicals in the afternoon, a topic outside the scope of this manuscript."

We have inserted "and section 3.1.3." following Figures 3 and 4 since the timing of the ozone maximum is discussed in that section.

**Section 3.1.5: This section needs the most revision, particularly with respect to the presentation and discussion of the ACSM data. The authors quantify fractions of "total aerosol mass" (e.g. line 541); however, only non-refractory submicron aerosol was measured. It is expected at refractory sea salt aerosol contributes significantly to this site, so these calculations are expected to be inaccurate.**

The paragraph starts out with "The ACSM submicron aerosol composition data ..." to acknowledge this important point. Furthermore, on line 540 we specifically state "... mass fraction of the non-refractory aerosol". However, we agree that the term "total aerosol mass" could have nevertheless been misunderstood and have qualified "total aerosol mass" by adding "measured by the ACSM" on line 541. We also inserted "$PM_1$" throughout the manuscript for additional clarity that larger particles were not quantified.

Since we report fractions within the non-refractory submicron aerosol, they are accurate (within the ability of the ACSM to make such measurements), not inaccurate as the reviewer claims.

*Similarly, the authors discuss the "inorganic mass fraction" and "most abundant inorganic component", which also are influenced by refractory aerosol, such that the mass fractions are expected to be inaccurate.*

We believe it is obvious from the context (especially after the changes already made to the comments above) that we refer to the ACSM data, i.e., non-refractory aerosol. We disagree with the reviewer's assertion of inaccuracy and note that the mass fractions are accurate within the ability of the ACSM to make such measurements. No further modifications were made to the manuscript in response to the above comment.

*The discussion of the ACSM data must clearly reflect that only submicron non-refractory mass was measured*

This is stated on lines 343-344 ("The chemical composition of non-refractory PM$_1$ was monitored using an Aerosol Chemical Speciation Monitor (ACSM, Aerodyne)") in the experimental section.

No further changes were made to the manuscript in response to the above.

*and that sea salt aerosol (most relevant for ClNO2 production) was not measured.*

This is stated in the abstract on lines 36-37 "unquantified supermicron sized or refractory sea salt derived aerosol" but had not been reiterated in the text. We added the following on line 346:

"The composition of the refractory aerosol (i.e., sea salt) was not quantified."

*Lines 509-517: It should be clarified that this only reflects the aerosol <0.5 um in diameter, based on the size range measured by the SMPS. Since aerosol surface area peaks at a higher diameter than aerosol number, it would be expected that this calculation of aerosol surface area may be a significant underestimate. This should be stated, and the implications of this should be discussed where appropriate.*

The reviewer is probably referring to the section titled "aerosol size distribution measurements" (section 3.1.5) that appears on lines 527-536.

The size range of the SMPS was already stated on line 348 (10 to 487 nm).

We question the reviewer's assertion that the surface area is significantly underestimated. The aerosol in the LFV is of urban and rural organic nature (see, for example, the title of (Alfarra et al., 2004)). John Seinfeld and Spiros Pandis state in their third and most recent edition of "Atmospheric Chemistry and Physics: From air pollution to climate change" in section 8.2.1 "Urban aerosols" on pages 343-344 that "... most of the surface area is in the 0.1 to 0.5 μm size range".

Below, we have included a graph of median aerosol surface area distribution observed in this study. The graph shows that the bulk of the surface area is captured.

[Figure]

We added the following on line 531:

"The size distribution data show that bulk of the surface area (i.e., the mean diameter $(\overline{D}_S)$) is in the range of 200 to 300 nm, such that most of the area of the accumulation mode was captured. However, the surface area calculations do not include contributions from larger diameter particles which were not quantified."

*Lines 554-558: The authors should refer to Zhang et al (2007, Environ. Sci. Technol.) for the proper method for examining aerosol acidity using ACSM data.*

Zhang et al. give the following expression for neutralization ratio:

$$NH_4^+{}_{meas}/NH_4^+{}_{neu} = NH_4^+/18/(2 \times SO_4^{2-}/96 + NO_3^-/62 + Cl^-/35.5) \quad (2)$$

In our expression, the square brackets denote molar concentrations, which is now clearly stated in the text. We converted the mass concentrations (micrograms per cubic meter) to molar concentrations (molecules per cubic centimeter) using the appropriate molecular weights (18 g/mol, 96 g/mol, and 62 g/mol etc.). Inclusion of the numbers 18, 96, 62, etc. in the equation is only necessary if one normally works with mass concentrations. We omitted the chloride concentrations because they were negligible.

However, there was a mistake in our equation in that the stoichiometric factor of 2 was left off by mistake. This has been corrected, and the median NR value was recalculated to 1.19. We also added a citation to Zhang et al., 1997 for the NR calculation.

The sentence in question now reads:

"The neutralization ratio, NR ≈ $[NH_4^+]$:($[NO_3^-]$+2$[SO_4^{2-}]$) (Zhang et al., 2007),  where the square brackets denote molar concentrations (calculated from the mass concentrations reported by the ACSM by dividing by the appropriate molecular weights), was 1.19 (median value)."

We have also updated Figure 6C to reflect the corrected NR values. The changes were sufficiently minor that the text did not require further revision.

*Please show the gas-phase NH3 data, at least in the supplementary information.*

Done (See page 2 above).

***Lines 559-562 and Figure 6 caption: Only non-refractory chloride was monitored by the ACSM, and this should be noted, given the importance for ClNO2 production.***

We inserted "non-refractory" prior to "chloride" as per the reviewer's request.

***If the signal was below the instrument limit of detection, then the concentration calculated is, by definition, not accurate.***

I believe we're in agreement here - we stated on line 560 (relevant text passage underlined):

"The ACSM software also identified reported non-refractory chloride with an average (±1 standard deviation) concentration of 0.01±0.03 µg m$^{-3}$, though it is unclear if this signal was real as it did not vary over the course of the campaign and was below the stated ACSM detection of limit of 0.2 µg m$^{-3}$ (Ng et al., 2011)."

No further changes were made to the manuscript.

***The text should be revised to reflect these two important points.***

Done.

***Lines 566-570: The time of year is expected to be quite important for these comparisons. Please provide this information in the discussion.***

Both studies were conducted in the summer: Pacific 2001 was conducted in August, 2001, which is now stated in the text:
"Previous AMS measurements in Vancouver during the month of August as part of Pacific 2001 ..."

***Lines 616 & 621: Do these dry deposition rates make sense in the context of previously published literature?***

Yes, they do. For dry deposition of $O_3$ and $NO_2$, as stated in section 2.5, we used dry deposition rates from Lin et al. Atmos. Environm., 44, 4364-4371, 10.1016/j.atmosenv.2010.07.053, 2010 and ran several simulations, varying mixing height.

For dry deposition of $NO_3$ and $N_2O_5$ it is less clear as there have been few reports let alone measurements of deposition velocities. For example, Kim et al. [PNAS, 2014] reported $N_2O_5$ exchange velocities of -1.7+/-0.6 cm/s with an ocean surface, and Bill Simpson studied uptake of $N_2O_5$ on snow in Alaska. The surface of the LFV is covered by vegetation and is obviously quite different from marine and snow covered environments. We are not aware of flux measurements of $N_2O_5$ or $NO_3$ on such terrain, but believe the magnitude assumed here to be feasible given the reactive nature of $N_2O_5$ and $NO_3$.

*Lines 629-630: Why is this data not shown? It is about modeling the "faster Ox loss at the beginning of the night", which seems central to the section header "Box model simulations of the nocturnal O3 and Ox loss in the NBL".*

As stated above, we have added this result as Figure S-3 (see pg. 4 above).

*Lines 632-634: Why was the model not simply constrained to measured NO?*

Any NO emitted will react with $O_3$ and hence "add" $NO_2$. Simply constraining NO to measured quantities is unlikely to capture this effect. No changes were made in response to this comment.

*Did NO under these modeled conditions match what was measured?*

Yes, they do (see figure below). The black line is the amount of NO the model predicts; the purple area is the 75%-25% percentile of observed (solid line is the median).

[Figure]

The figure above has been added to the S.I. as Figure S-4.

The main manuscript text was changed as follows:

"There is reasonable agreement between the simulations and observations of $O_x$ and $O_3$ until ~3:00 (and between simulation and observation of NO, Figure S-4). This, which shows that the nocturnal $O_3$ and $O_x$ loss can be rationalized without active $NO_3$ and $N_2O_5$ chemistry and suggests that $NO_3$, $N_2O_5$, and $ClNO_2$ did not contribute significantly to $O_x$ and $O_3$ loss in the NBL."

*Lines 692-694: Not including loss of NO3 to hydrocarbons is not justified here. Why not assume a generic BVOC as you did in an earlier section, or use the data that you do have? These options seem better than blindly ignoring this NO3 loss process.*

*Further, on lines 706-707, you state that reaction between NO3 and isoprene was likely significant, suggesting that that reaction should be included, especially since the authors have at least some measurements of isoprene.*

We neglected the reaction of $NO_3$ with unsaturated VOCs because there simply wasn't sufficient data to include it: As stated in the manuscript "Missing from equation (3) are losses of $NO_3$ to hydrocarbons (which was omitted because of the poor VOC data coverage)", VOC data were only available for the nights shown in Figure 7. On those nights, unfortunately, mixing ratios of $O_3$ and hence $N_2O_5$ and $ClNO_2$ were small, and any isoprene concentrations measured would have been larger than on other nights (due to the difference in the magnitude of chemical sinks) so those VOC data cannot be simply extrapolated to other nights. We clearly stated that the results therefore are upper limits only.

On the other hand, not including reactions of $NO_3$ with VOCs gave us an important result: Figure 8C shows that the experimental $N_2O_5$ lifetime from 21:00 to midnight was lower than calculated from the expression given on line 691. Part of this may be due to the time of ~70 min it takes to reach steady state. The remainder is likely due to $NO_3$ reacting with residual isoprene or terpenes that continue to be emitted (stated on lines 706-707). The effect of these VOCs appears to lessen over the course of the night.

No changes were made in response to the above comment.

*Also, not including NO3 and N2O5 deposition is not justified. It seems that dry deposition could be easily included.*

This is correct and was an oversight on our part. We have revised the manuscript, added dry deposition ($k_{dep}(NO_3)$ and $k_{dep}(N_2O_5)$) to the lifetime expression and recalculated the upper limit to the $N_2O_5$ lifetime. Earlier in the text (section 3.2.1), we estimated that $k_{dep}(NO_3)$ and $k_{dep}(N_2O_5)$ were ~$4\times10^{-4}$ $s^{-1}$ at night; during the day, these are, of course, much smaller due to higher mixing heights (though this won't matter since the daytime sinks for $NO_3$ and $N_2O_5$ are much larger).

The paragraph in question now reads as follows:

" Superimposed on the right-hand side of Figure 8C are upper limits to the steady state lifetime of $N_2O_5$, calculated using the sum of pseudo first-order rate coefficients for the titration of $NO_3$ by NO ($k_3$[NO], R3), $NO_3$ photolysis ($j(NO_3)$, R4), and $NO_3$ dry deposition ($k_{dep}(NO_3)$), all divided by the $N_2O_5$ over $NO_3$ ratio at equilibrium given by $K_2NO_2$ (Figure 8B), plus the pseudo first-order rate coefficient for $N_2O_5$ heterogeneous uptake ($k_{het}(N_2O_5)$, equation (1)) plus $N_2O_5$ dry deposition ($k_{dep}(N_2O_5)$).

$$\tau(N_2O_5) = \left(\frac{k_{NO_3}}{K_2[NO_2]} + k_{N_2O_5}\right)^{-1} < \left(\frac{k_3[NO] + j(NO_3) + k_{dep}(NO_3)}{K_2[NO_2]} + k_{het}(N_2O_5) + k_{dep}(N_2O_5)\right)^{-1}$$

(3)

The dry deposition rate constants were set to $4 \times 10^{-4}$ s$^{-1}$ (see section 3.2.1), which likely overestimates dry deposition during the day due to higher mixing heights; however, the error this introduces is negligible compared to the large daytime sinks such as NO$_3$ photolysis and its reaction with NO. "

We also updated Figure 8C (see next comment).

*Where time periods that fog and rain occurred used? If so, these periods should be removed for these calculations.*

Please note the following modifications in response to the above comment:

"Missing from equation (3) are losses of NO$_3$ to hydrocarbons (which was omitted because of the poor VOC data coverage) and terms for NO$_3$ and N$_2$O$_5$  wet (i.e., on cloud and rain droplets) deposition. Periods affected by precipitation or fog (shown in Figure 3D) were hence excluded from the calculation."

Only a small fraction of the data set was affected by rain or fog. It therefore made little difference to the final result if those data are included or excluded in this data set. Likewise, the inclusion of the dry deposition terms only marginally changed the results of the calculations (the other sinks were just that much larger), such that discussion of these results did not need to be changed.

*Lines 721-723: It is assumed that the production of nitrate on refractory aerosol is minimal, but this is not justified and is a poor assumption. Nitric acid displacement of HCl is one of the most common sea salt aerosol aging pathways (Gard et al. 1998, Science).*

We agree with the reviewer and have clarified the sentence as follows:

"It is assumed further that production of nitrate from N$_2$O$_5$ uptake on refractory aerosol (that the  ACSM does not quantify) is minimal."

*Line 757-762: Only non-refractory chloride and nitrate were measured! This should be considered and reflected in this discussion.*

This is true, but it is entirely possible that uptake of N$_2$O$_5$ on refractory aerosol and non-refractory are different in terms of kinetics and ClNO$_2$ yield (the Bertram and Thornton groups have shown this). We explicitly state this is a condition ("if one assumes that all of the ClNO$_2$ is produced on supermicron or refractory aerosol such that P(ClNO$_2$) on submicron aerosol equals 0 pptv s$^{-1}$"). The nocturnal uptake of N$_2$O$_5$ (whose mixing ratios were measured) has been shown again and again to be an important source of particle-phase nitrate on non-refractory aerosol (whose concentrations were monitored by the ACSM). In this paper, we can make statements about the processes on non-refractory aerosol.

*Line 827: How does this observation of a lack of non-refractory submicron aerosol chloride compare to other similar inland coastal AMS/ACSM studies?*

Alfarra et al. and Boudries et al. (Alfarra et al., 2004; Boudries et al., 2004) reported AMS data collected during PACFIC 2001 but did not report aerosol chloride as one of their products.

There were, however, size-resolved MOUDI measurements during PACIFIC 2001 (Anlauf et al., 2006). We have modified the text as follows:

"This in turn implies that the submicron aerosol surface did not significantly participate in the production of $ClNO_2$ from $N_2O_5$ uptake in the NBL, broadly consistent with the conclusions in section 3.2.3 and consistent with measurements of water-soluble aerosol components in the LFV during Pacific 2001 (Anlauf et al., 2006) that showed no evidence for chloride redistribution to $PM_1$ from larger particles where aerosol chloride was present."

*Lines 913-916: The authors suggest that the lack of particle-phase chloride (should be 'non-refractory' chloride) is in contrast to their previous study, Mielke et al. 2013, in Pasadena, CA. However, in that previous paper, AMS PM1 showed very little nonrefractory chloride, with far higher levels of PM2.5 chloride (refractory + non-refractory measured by PILS-IC) measured. So, this is not a complete comparison, and in fact, in terms of non-refractory PM1 chloride, the studies seem fairly similar. This discussion should be reconsidered and revised.*

We disagree. In the 2010 study, a median value of ~0.1 $\mu g\ m^{-3}$ aerosol chloride were observed by the AMS on the non-refractory aerosol fraction, ~0.1 $\mu g\ m^{-3}$ more than that was observed in this study. So yes, there was a sizeable difference. We do not show how $PM_{2.5}$ chloride compares since we don't have such data for the Abbotsford study; those levels are likely "far higher" at both locations.

We have modified the text on line 830:

"Such a redistribution was observed, for example, during the Calnex-LA campaign, where the AMS measured a median chloride concentration of ~0.1 $\mu g\ m^{-3}$ on non-refractory aerosol (Mielke et al., 2013)."

*Table 1: Please add the SMPS and ACSM size ranges, as well as note that the ACSM aerosol composition reflects only the "non-refractory" aerosol.*

These details are stated in the experimental section. No further changes were made to manuscript.

*Minor Comments:*

*Line 65: Fix typo – "particle" should be "particulate".*

Done

*Reaction 6 should include chloride as a reactant.*

Added

*Please add references to the following lines: 77, 99, 108, 426, 431, 887.*

Done, with exception of lines 426 and 431 since it is common knowledge in the field that NO is emitted by automobiles.

*Lines 135-136, 837: Fix reference formatting in sentence.*

Done

*Line 337: Fix typo – "day used" should likely be "day were used".*

Done

*Section 3.4, Table 2, & anywhere else: "down-dwelling" and "up-dwelling" should be "down-welling" and "up-welling".*

Done

*Line 457 & elsewhere: Error should be given as one significant figure, with the average value provided with the same number of decimal places. For example, line 457 should list 64 +/- 1 ppbv, rather than 64.4 +/- 1.2 ppbv.*

Done

*Lines 459-461: Provide values in parentheses for context.*

We changed the text as follows:

"These levels were well below the CAAQS 8-hr standard of 63 ppbv and the 1 hour National Ambient Air Quality Objective of 82 ppbv, smaller than the pre-2003 data analyzed by Ainslie and Steyn (2007), who reported between 10 and 20 $O_3$ 1-hour exceedences of 82 ppbv in the 1980s, and of similar magnitude as observed by a high-density monitoring network in the region in 2012 (Bart et al., 2014), which observed peak $O_3$ levels of 74 and 83 ppbv at Abbotsford on July 8 and August 17, respectively."

*Line 467: Change ":" to "."*

Done

***Line 469: Fix typo – "loss of are" should likely be "loss are".***

Done

***Line 482: Fix typo – "at a median value" should be "at median values".***

Done

***Lines 483-484: Fix typo – "ratio of this campaign was" should be "ratios of this campaign were".***

Done

***Figure 5: Please clarify this figure caption. It was not obvious at first what "lhs" and "rhs" stood for, as these acronyms are not defined.***

Done

***Lines 601-603: Please revise sentence to improve clarity.***

Original sentence:

"Loss of $NO_3$ to isoprene was a small sink compared to its loss to NO via reaction (3) and $NO_3$ photolysis but is approximately on par with $k_{N2O5}$ $K_2[NO_2]$."

Revised sentence:

"Loss of $NO_3$ to isoprene was a small sink compared to its loss to NO via R3 and $NO_3$ photolysis (R4) but was approximately on par with its indirect loss, i.e., the heterogeneous uptake of $N_2O_5$."

***Lines 854-859: Please revise sentence to improve clarity.***

Original sentence:

"Their presence is likely responsible for some of the gap between the low "observed" $N_2O_5$ steady state lifetimes, $\tau(N_2O_5)$, compared to the upper limit set by reactions 3-4."

Revised sentence:

"Their presence is likely responsible for the difference between the "observed" $N_2O_5$ steady state lifetimes, $\tau(N_2O_5)$,  and upper limit calculated using equation (3) before midnight (Figure 8C)."

*Line 658: Fix typo - "are" should be "is".*

Done

*Line 664: Delete sentence as this information is already given on line 660.*

Done

*Lines 730 & 914: Fix typo – "ACMS" should be "ACSM".*

Done

*Anonymous Referee #2

*This manuscript reports measurements of ClNO2, N2O5 and other chemicals (ozone, NOx, NOy, aerosol size and composition and VOCs etc) at a surface site near the Lower Fraser Valley during a two-week period in summer 2012. The study was motivated by the need to investigate the role of ClNO2 in ozone exceedance in the region.*

*However, the relatively short field study did not capture any high ozone events, and low ClNO2 levels were observed due to fresh emissions of NO which suppress the production of N2O5 and ClNO2 at nigjt. The paper investigated some metrics related to production and loss of N2O5/ClNO2 with the aid of a simple box model, and the results show small contribution of ClNO2 to radical production in such an environment, as one would expect.*

*While the data on ClNO2 and N2O5 add to the global data base of the two important and poorly documented species, the main finding (low N2O5/ClNO2 in high NO condition and resulting small contribution of photolysis of ClNO2 to radical source) gives limited new insight on the processes and impact of N2O5 and ClNO2, as such the significance of this work is unclear.*

We thank the reviewer for these comments. The reviewer states that "the study was motivated by the need to investigate the role of $ClNO_2$ in ozone exceedance in the region". While we agree that this is a penultimate goal, the focus of this paper was to provide observational data, as there had been no prior measurements of $ClNO_2$ and $N_2O_5$ mixing ratios in the LFV; in this work, we presented the first such measurements, which are challenging in their own right, and provided an analysis of nitrogen oxide budgets, nocturnal $O_3$ loss, and a comparison of photochemical radical sources.

The mixing ratios of $ClNO_2$ at ground level were low, in fact, considerably lower than current literature on $ClNO_2$ suggests (Table 3). However, there is general bias in how measurement sites have been selected (and are reported): They're typically located in highly polluted regions (such as the LFV), and as a consequence, many $ClNO_2$ data sets to date reported fairly large mixing ratios (many ppbv) that

constitute large fractions of $NO_y$. Not in this work – in spite of the proximity to pollution sources ($NO_x$ from a Megacity) and the Pacific Ocean which provides sea salt aerosol.

Could this have been expected? Perhaps. Though doubtful.

A key aspect of this data set is the lack of redistribution of aerosol chloride from the refractory sea salt derived aerosol to the refractory fraction. This suggests that uptake of $H_2SO_4$ and $HNO_3$ on sea salt aerosol and displacement of HCl were less active than at other locations (e.g., Pasadena). This is surprising insofar as acid displacement, chloride deficits, etc. are well documented. Since the aerosol surface in this study is dominated by the smaller, non-refractory size fraction, most of $N_2O_5$ reacts on that surface. The lack of chloride implies the efficiency of $N_2O_5$ to $ClNO_2$ conversion on that surface is also much lower.

We believe that this paper adds to the body of work on $ClNO_2$ and nitrogen oxide chemistry in general in that the data allow constraints to be placed on future photochemical models of $O_3$ production in the region. Currently, we are not aware of any chemical transport model simulations of $O_3$ production in the LFV that incorporate chlorine chemistry. In addition, this work provides valuable guidance for future field campaigns in the region in that the focus should be on processes happening within the residual layer and measurements of aerosol composition should include refractory aerosol.

***Specific questions on methods:***

***What was the extent of N2O5 loss in the sampling line during the field study?***

This information was stated on line 297
" The $N_2O_5$ response varied between 65% and 100% depending on inlet "age"; the Teflon™ inlet and aerosol inlet filter were changed every 2 – 3 days."

Please note that this response factor also accounts for laser wavelength drifts from the $NO_3$ absorption line (likely less variable than $N_2O_5$ losses in the sampling line).

We have modified the sentence slightly to reflect this:

"The $N_2O_5$ response (which accounted for $N_2O_5$ loss in the sampling line and slight mismatches of the laser wavelengths with the $NO_3$ absorption line) varied between 65% and 100% and depended on inlet "age""

***For NOy measurements, was the Mo converter placed at the sample inlet outside?***

Yes, it was placed outside. The following was inserted into the manuscript (on line 307)

"An $NO$-$O_3$ chemiluminescence instrument (Thermo 42i) was used to monitor mixing ratios of NO and $NO_y$, which was reduced to NO in a Mo converter heated to ~320 °C placed outside a short distance (< 1 m) from the sample inlet."

*Was a filter placed before the Mo converter?*

Yes, there was a filter. This was stated on line 308:

"This instrument sampled from the main inlet via a Teflon™ filter and filter holder ...."

*The aerosol size measurements only covered size 10 nm to 487 nm, were the larger size particles considered when calculating the aerosol surface areas density?*

Larger size fraction data were not available, so this was unfortunately not possible. Since the limitations and consequences are already noted in the text, not changes were made to the manuscript in response to this comment.

*What was the uncertainty of the simple box model adopted?*

Uncertainty in box models are generally difficult to assess accurately. This is typically accomplished by probing the sensitivity to changing inputs. Here, we ran several simulations. The data in Figure S-1 show that changing the dry deposition rate of $O_3$ by a factor of 2 causes a considerable deviation between model and observation. Thus, the model is at least accurate within that factor.

This, of course, hinges on whether the assumptions in this simple model hold, such as a negligible role of nighttime nitrogen oxide chemistry in $O_3$ and $NO_2$ depletion. This particular assumption is quite reasonable as the model simulations to assess the time to steady state (which neglect dry deposition) show that $NO_3$/$N_2O_5$ chemistry at most only removes ~3 ppbv of $O_x$.

*Anonymous Referee #3

*General Comments:*

*This is a well written manuscript describing studies of nocturnal chemistry in the Lower Frasier Valley, a region near a megacity (Vancouver) and with sea salt sources. The combination of these pollution (NOx) sources and sea salt aerosol particles might be expected to produce nitryl chloride. In fact, Pasadena, CA, in the Los Angeles area (also a megacity near the ocean) had much larger nitryl chloride at ground level. The authors argue that shallow boundary layers, titration of ozone by fresh NO emissions at ground level and potentially biogenic VOC inputs often preclude nitryl chloride formation at ground level, which is supported by the data in the manuscript. The work is well written, sufficiently referenced, and appropriate for publication in ACP.*

We thank the referee for this positive opinion of the manuscript.

*There is a lot of evidence in this manuscript that titration of ozone at ground level was a reason for low NO3 and N2O5 levels. On the nights when there was ozone, N2O5 and ClNO2 were at their largest mixing ratios. Presumably this titration is caused by input of NO at ground level, which does not mix to very high altitude. Therefore, it is likely that aloft there is more active N2O5 chemistry and probably ClNO2 production. For this reason, I think that the manuscript's title should be modified to include "Low levels of nitryl chloride at ground level...".*

An excellent suggestion. We have modified the title as suggested by the referee.

*The peaking of ClNO2 after sunrise and coincident with breakup of the nocturnal boundary layer would seem to indicate that ClNO2 aloft is likely higher.*

Agreed.

*The manuscript has discussion about chloride measurements based upon the ACSM, which is not good at detecting chloride in the form of NaCl.*

This is correct. Please see our replies to referee #1 (who raised this issue as well).

*There should be measurements in the area of PM2.5 chemical composition that could help to better understand the presence of sea salt chloride. The authors should examine available aerosol chloride measurements to expand their analysis and interpretation through consideration of these data.*

We agree that lack of refractory aerosol composition measurements is a limitation of this study. This limitation is noted in section 3.2.3 on lines 751-752 "... production of $ClNO_2$ from uptake of $N_2O_5$ on unquantified supermicron (i.e., > 0.5 μm) or refractory aerosol (which takes place simultaneously) is not accounted for."

There are not many aerosol chloride data sets out there. Anlauf et al. showed that sea salt aerosol is present in the LFV, but mainly in the supermicron size fraction that was not quantified in this work. We are now citing Anlauf et al. (2006) on line 756 (since we showed supermicron aerosol in the LFV to be mainly sea salt derived, so we no longer need to speculate about that). Since we are already concluding that ClNO2 production that does take place happens mainly on unquantified sea salt aerosol, our conclusions do not need to be revised or altered.

We hope that the referees understand that in this line of work, we are sometimes compromised by budgetary constraints, and will not always have every measurement at our disposal, no matter how desirable such measurements might be. In response to the reviewer's comment we have modified the final sentence of the discussion section to express this desire:

"Future studies should therefore target the NRL, for example through missed-approaches by aircraft, a blimp, or from a tall tower, especially during episodes of a developing $O_3$ exceedance event and also include composition measurements of refractory aerosol."

**Specific comments:**

**Showing population density (in some way) on the Figure 1 map would be nice.**

We agree but since we don't have access to such data have chosen to leave Figure 1 as is.

**Figure 3 seems to be mentioned before Fig. 2**

We don't believe this to be the case. Figure 2 and Figure 3 are first mentioned on lines 234 and 386, respectively. On line 193, we call out Figure 3 of another paper, though. No changes were made to the manuscript.

**Line 194: I think the word is "aging"**

We agree and have made the change in the text.

**Line 199: I don't think ECCC is defined?**

This has been fixed.

"ECCC Environment and Climate Change Canada (ECCC)"

**Line 202: Define THS?**

It's a company name (named after Tanner, Huey and Stickel). We have inserted the word "Instruments" following "THS" to make that clearer.

*Line 360: Presumably after measuring the upwelling/downwelling actinic ratio, this ratio was used to correct all downwelling actinic flux data to be a total actinic flux. If this was done, it should be noted.*

This was noted on lines 359/360

"On several days, the spectrometer was inverted hourly to determine the up-welling radiation, which was added to the down-welling flux."

No changes were made to the manuscript in response to this comment.

*Line 394: I think K2 is first used here but not defined until later. This should probably be done near line 80*

We have added "In ambient air, $N_2O_5$, $NO_3$ and $NO_2$ are usually in equilibrium; the equilibrium constant, $K_2$, is temperature dependent, favoring $NO_3$ and $NO_2$ at higher temperatures (Osthoff et al., 2007)." near line 80 as requested.

*Line 533: This equation is applicable when diffusion limitations are not important, which might not be true if supermicron particle surface area is involved.*

An important point. We have added the sentence "Equation (1) is valid for uptake on small, submicron aerosol as it neglects gas-phase diffusion limitations (Davidovits et al., 2006).

*Line 554: Make clear that an equivalent basis (e.g. 2x the molar concentration of sulfate) is being used in this neutralization ratio equation.*

Fixed (issue already raised by reviewer #1).

*Line 559-562: Does the ACSM detect chloride efficiently? Standard filter samples would show chloride and could be used to verify its presence or absence. Historical data from the area would tell you the ratio of chloride to other inorganic ions (e.g. nitrate and sulfate), so you should be able to tell if the ACSM is not actually detecting chloride efficiently.*

This point was already raised by reviewer #1 – please see our responses above.
Briefly, the reviewers are correct in that the ACSM does not quantify "refractory" chloride (i.e., chloride present in sea salt derived aerosol). Unfortunately, no other data are available since filter samples were not collected during this campaign. For historical data, during PACIFIC 2001 there were AMS measurements (Alfarra et al., 2004; Boudries et al., 2004) and measurements of water-soluble aerosol components (Anlauf et al., 2006). All these papers are now cited.

***Line 751 area: Presumably some pollution monitoring studies looked at PM2.5 via IC and could address presence of at least 1 to 2.5 micron particles containing Cl-***

Correct. Anlauf and coworkers collected MOUDI data during PACIFIC 2001 and showed variable sea salt concentrations in the supermicron size fraction (Anlauf et al., 2006).
We have modified the following sentences in the introduction:

[revised manuscript text omitted]

---

## Author Response (AR2)

Hello Tim:

Your comments are reproduced below in **_bold italic font_**. Responses are given in regular font.

Cheers,

Hans

*Comments to the Author:*

*Dear Hans-*

*Thank you for submitting the revised manuscript. I think that you have answered the reviewers comments clearly and I appreciate the additional information that has been supplied to the SI.*

*A few quick comments from my end:*

*Line 735: While I appreciate that the VOC data coverage is sparse, I don't think this should be given as a reason for not including bounds for the impact that NO3-VOC reactions may have on the analysis. Is it possible to run a sensitivity test to assess how important VOCs might be in the N2O5 analysis if they were measured? For example, at what NO3 reactivity does this become an issue to the interpretation of the results.*

A hypothetical estimate as suggested is speculative but certainly possible. We inserted the following at the end of the paragraph in question (and also took the liberty to correct a grammatical error):

"Missing from equation (3) are losses of $NO_3$ to hydrocarbons (which  were omitted because of the poor VOC data coverage) and terms for $NO_3$ and $N_2O_5$ wet (i.e., on cloud and rain droplets) deposition. Periods affected by precipitation or fog (shown in Figure 3D) were hence excluded from the calculation. Estimates on how loss of $NO_3$ to VOCs could affect the lifetime of $N_2O_5$ are given in the S.I."

The following was added to the S.I.:

**Estimates of how loss of $NO_3$ to VOCs would affect the lifetime of $N_2O_5$**

The steady state lifetime calculation presented in Figure 8C of the main manuscript neglects losses of $NO_3$ to VOCs due to poor data coverage, i.e., presents a scenario where $\Sigma k_{NO3+VOC,i}[VOC]_i$ is assumed to be zero, which is, of course, unrealistic.

We used all available VOC data and calculated a time series of $\Sigma k_{NO3+VOC,i}[VOC]_i$. The average (±1 σ) value at night is $(0.038\pm0.026)$ $s^{-1}$. The $N_2O_5$ loss frequency, calculated by dividing this value with the $N_2O_5$:$NO_3$ ratio, is $(1.1\pm0.9)\times10^{-5}$ $s^{-1}$, corresponding to a lifetime of ~2.5 hours, which is negligible.

However, as stated in the main manuscript, the VOC data coverage is sparse and did not include measurements of all hydrocarbons towards which $NO_3$ is reactive. Recently, Liebmann et al. (2018) reported an average of nocturnal $NO_3$ loss frequency of 0.11 $s^{-1}$ in the boreal forest of Finland. This value likely included loss of $NO_3$ to NO and a variety of hydrocarbons such as sesqui- and diterpenes, which are likely present in higher concentration in a boreal forest than at Abbotsford and hence represents an upper limit Taking this value and multiplying it by the $N_2O_5$:$NO_3$ ratio, the average nocturnal $N_2O_5$ loss frequency via $NO_3$-VOC reactions is calculated to $(5.6\pm1.3)\times10^{-3}$ s$^{-1}$. Figure S-8 shows the result of including this value in the calculation of $N_2O_5$ lifetime.

[Figure]

— $((0.11$ s$^{-1}+j(NO_3)+k_3[NO]+k_{dep}(NO_3))/K_2[NO_2])+k_{het}(N_2O_5)+k_{dep}(N_2O_5))^{-1}$
— $\tau(N_2O_5)$

**Figure S-8**. Same as Figure 8c but including an assumed $NO_3$ loss frequency to VOCs of 0.11 s$^{-1}$.

The paragraph following Figure 8 in the main manuscript was expanded as follows:

"The median "observed" $\tau(N_2O_5)$ is below or equal to the upper limit calculation with equation (3) during both night and day. The largest discrepancy is observed at the beginning of the night, when oxidation of (unsaturated) hydrocarbons by $NO_3$ (R7) was likely most significant due to the presence of isoprene and other biogenic VOCs. Indeed, if the $\Sigma k_{NO3+VOC,i}[VOC]_i$ is assumed to be 0.11 s$^{-1}$ (average nocturnal $NO_3$ loss frequency reported by (Liebmann et al., 2018)), the gap between observed and calculated $N_2O_5$ lifetime between sunset and midnight closes (Figure S-8). However,  this is also the time when the steady state approximation is most likely invalid."

We are also referring to Figure 8C in the discussion (around line 900):

"Emissions of monoterpenes, which are reactive towards $NO_3$, are driven by a temperature-dependent process from storage tissue within the plants at night (Guenther et al., 1995) and, hence, were likely substantial. Their presence is likely responsible for the difference between the "observed" $N_2O_5$ steady lifetimes, $\tau(N_2O_5)$, and upper limit calculated using equation (3) before midnight (Figures 8C and S-8)."

**Deposition rates: Are the deposition velocities that are compared to (for O3 and NO2) made at night?**

Yes. We used deposition velocities from Lin et al. (2010). Their paper is titled "Simple model for estimating dry deposition velocity of ozone and its destruction in a polluted nocturnal boundary layer". It is hence safe to assume that their values are based on measurements made at night.

*I am somewhat surprised that deposition occurs this rapidly at night for this region.*

We were surprised as well, and that's exactly why we show the calculation and spent a lot of space in the paper on this. It seems to come down to mixing height – the shallower, the faster the loss.

*I would think the same would be true for NO3 and N2O5. The N2O5 deposition velocities from Kim et al (over water) are most likely sustained by shear induced turbulence. I would expect this to go to near zero at night over land in the absence of convectively driven turbulence. How would this impact your analysis.*

Good point. One of the advantages of the deposition velocity representation is that all the complexities of the dry deposition process are bundled in a single parameter, $v_d$, though that shouldn't be a licence to ignore such complexities.

A few paraphrased thoughts from Seinfeld and Pandis to frame this discussion: It is generally assumed that transport of material to the surface is governed by three resistances in series: the aerodynamic resistance ($r_a$), quasi-laminar layer resistance ($r_b$), and canopy resistance ($r_c$). The first two terms are affected by wind speed and atmospheric stability. Higher resistances are expected under stable and neutral conditions, and under stable conditions, especially with low winds, $r_a$ dominates dry deposition. At night $r_a$ is considerably higher over land and is often the controlling factor in the overall rate of deposition. $r_a$ values up to 150 s/m can occur at night over land when turbulent mixing is reduced.

In this data set, the local wind speeds (Figure 3D) were lower during the night than during the day, about 1 m/s on average. This would imply a large aerodynamic resistance and that $v_d$ ($\approx 1/r_a$) is in reality larger than the value taken from Lin et al. (2010) that was assumed for the analysis.

We have added the following to the first paragraph of section 3.2.1:

[revised manuscript text omitted]

Ethene, 1,2-dichloro-, (E)-

Methyl tertbutylether (Propane, 2-methox

Ethane, 1,1-dichloro-

Vinyl Acetate (Acetic acid ethenyl ester

2-Butanone

Chloroform (Trichloromethane)

Ethyl Acetate

Furan, tetrahydro-

Ethane, 1,2-dichloro-

Ethane, 1,1,1-trichloro-

Carbon Tetrachloride

Trichloroethylene

Methane, bromodichloro-

1,4-Dioxane

Methyl Methacrylate

1-Propene, 1,3-dichloro-, (Z)-

Methyl Isobutyl Ketone

1-Propene, 1,3-dichloro-, (E)-

Ethane, 1,1,2-trichloro-

2-Hexanone

Methane, dibromochloro-

Ethane, 1,2-dibromo-

Tetrachloroethylene

Benzene, chloro-

Bromoform (Methane, tribromo-)

Ethane, 1,1,2,2-tetrachloro-

Ethane, pentachloro-

Benzyl Chloride

Benzene, 1,3-dichloro-

Benzene, 1,4-dichloro-

Benzene, 1,2-dichloro-

Benzene, 1,2,4-trichloro-

1,3-Butadiene, 1,1,2,3,4,4-hexachloro-

Naphthalene

Freon 12

Chloromethane

Freon 114

Vinyl chloride

1,3 Butadiene

Bromomethane

Chloroethane

Ethanol

1R-alpha-Pinene

Camphene beta-Pinene

D-Limonene

[Figure]

**Figure S-1**. Time series of gas-phase ammonia data reported by Metro Vancouver. Data were not quality-assured and are non-quantitative.

**Box model to rationalize $O_x$ loss by dry deposition**

A box model was set up to simulate the median nocturnal decays of $O_3$ and $O_x$. These simulations are intended as back-of-the-envelope type estimates of major processes only since an accurate description of the nocturnal boundary layer chemistry would require modeling of horizontal and vertical transport, i.e., altitude-resolved information (Geyer and Stutz, 2004).

Such information was not available in this work.

The reactions used in this model are summarized in Table S-2. The mechanism consists of $O_3$

and $NO_2$ dry deposition, titration of NO with $O_3$ (R8) and chemical loss of $O_3$ to a generic biogenic hydrocarbon. For dry deposition, the velocities of $v_d(O_3) = 0.2$ cm s$^{-1}$ and $v_d(NO_2) =$

$\alpha \times v_d(O_3)$ with $\alpha=0.65$ from Lin et al. (2010) were used. The rate constants for reaction with the generic biogenic hydrocarbon was set to that of $\alpha$-pinene with $O_3$ ($5\times10^{-11}$ cm$^3$ molec.$^{-1}$ s$^{-1}$, (Seinfeld and Pandis, 2006)).

Model simulations were carried out using a custom differential equation integrator macro in the software package Igor Pro (Wavemetrics) and were initiated with the campaign median $NO_2$

and $O_3$ concentrations observed at sunset.

**Table S-2**. Reactions included in box model to estimate dry deposition velocities

| Reaction | Rate constant |
|---|---|
| $O_3 \rightarrow$ products | $k_{dep}(O_3)$ |
| $NO_2 \rightarrow$ products | $k_{dep}(NO_2)$ |
| $O_3 + NO \rightarrow NO_2 + O_2$ | $4.8\times10^{-4}$ ppbv$^{-1}$ s$^{-1}$ |
| $O_3 + VOC \rightarrow$ products | 1.25 ppbv$^{-1}$ s$^{-1}$ |

[Figure]

Figure S-2. Observed and simulated $O_x$ loss in the NBL at Abbotsford assuming $O_3$ dry deposition rates of $2\times10^{-4}$ s$^{-1}$, $4\times10^{-5}$ s$^{-1}$, $2\times10^{-5}$ s$^{-1}$ and $1\times10^{-5}$ s$^{-1}$, corresponding to approximate mixing heights of 10 m, 50 m, 100 m, and 200 m.

[Figure]

Figure S-3. Effect of biogenic VOC emissions on $O_x$. The observed and simulated $O_x$ loss in the NBL at Abbotsford assuming an $O_3$ dry deposition rate of $4\times10^{-5}$ s$^{-1}$ are shown as green and blue traces, respectively. The red trace shows the effect of adding 1 ppbv of reactive biogenic VOC at sunset and continuous biogenic VOC emissions of $3\times10^5$ molecules cm$^{-3}$ s$^{-1}$ throughout the night.

[Figure]

**Figure S-4**. Comparison of observed and simulated NO mixing ratios after constant emissions of $2.9 \times 10^{-4}$ ppbv s$^{-1}$ (~1.05 ppbv hr$^{-1}$) of NO and $3 \times 10^{-5}$ ppbv s$^{-1}$ (~0.05 ppbv hr$^{-1}$) of NO$_2$ were added.

**Box model to determine the time necessary for $NO_3$ and $N_2O_5$ to achieve a steady state with respect to production and loss**

The validity of the steady state assumption was evaluated in a similar fashion as described by Brown et al. (2003) using a simple box model. Reactions and rate coefficients included in these simulations are listed in Table S-3. Model simulations were carried out using a custom differential equation integrator macro in the software package Igor Pro (Wavemetrics). Rate coefficients were calculated for a temperature of 286 K, which is the median nocturnal temperature of this study (Figure 8B). Simulations were initiated with the median nocturnal $NO_2$ and $O_3$ mixing ratios of 7.5 ppbv ($1.92 \times 10^{11}$ molecules cm$^{-3}$) and of either 18 ppbv ($4.5 \times 10^{11}$ molecules cm$^{-3}$) or 5.0 ppbv ($1.3 \times 10^{11}$ molecules cm$^{-3}$), respectively. The simulations assume pseudo-first order $N_2O_5$ and $NO_3$ loss with frequencies of $1 \times 10^{-3}$ s$^{-1}$ and between $1 \times 10^{-2}$ s$^{-1}$ and 0 s$^{-1}$, respectively.

Simulated temporal profiles of $NO_3$ and $N_2O_5$ are show in Figure S-5 (left axis) and those of $O_3$ and NO2 on the right axis. The subpanels A, B, and C are simulations with $k_{NO3} = 0$ s$^{-1}$, $1 \times 10^{-3}$ s$^{-1}$ or $1 \times 10^{-2}$ s$^{-1}$, respectively. In each case, the rate of change of $[N_2O_5]$ with respect to time, $d[N_2O_5]/dt$, approaches zero after a period of ~70 min, or less, indicating the time to approach steady state. The simulations also show that the amount of $O_3$ and $NO_2$ removed through chemical reactions of $NO_3$ and $N_2O_5$ are ~1 ppbv and between ~1.9 and ~1.6 ppbv over a period of 4 hours. These are upper limits as in this study much of the $NO_3$ was titrated by NO. In any case, loss of $O_3$ through nocturnal gas-phase is predicted to be rather small compared to the total $O_3$ loss observed (~26 ppbv over 9 hours, see section 3.1.3 and Figure 4C in the main text). Brown et al. (2003) show that in these scenarios, $NO_3$, $N_2O_5$, and $NO_2$ remain in equilibrium almost throughout; for completeness, the corresponding plot for these simulations is shown in Figure S-6.

As shown in equation (2) of the manuscript, the steady state lifetime is approximately equal to:

$$\frac{[N_2O_5]}{k_1[NO_2][O_3]} \approx \left( k_{N_2O_5} + \frac{k_{NO_3}}{K_2[NO_2]} \right)^{-1} \tag{2}$$

A comparison of these two expressions is shown in Figure S-7. The time when these two expressions are equal is equal to the time to steady state.

**Table S-3**. Reactions included in the box model to estimate the time for $NO_3$ and $N_2O_5$ to achieve steady state with respect to their production and loss

| # | Reaction | Rate coefficient |
|---|----------|------------------|
| R1 | $NO_2 + O_3 \rightarrow NO_3 + O_2$ | $2.28 \times 10^{-17}$ cm$^3$ molecule$^{-1}$ s$^{-1}$ |
| R2$_f$ | $NO_3 + NO_2 \rightarrow N_2O_5$ | $1.35 \times 10^{-12}$ cm$^3$ molecule$^{-1}$ s$^{-1}$ |
| R2$_r$ | $N_2O_5 \rightarrow NO_3 + NO_2$ | $0.00923$ s$^{-1}$ |
| (R7) | $NO_3 \rightarrow$ products | $k_x = k_{NO3} = 0$ s$^{-1}$, $1 \times 10^{-3}$ s$^{-1}$ or $1 \times 10^{-2}$ s$^{-1}$ |
| (R5) | $N_2O_5 \rightarrow$ products | $k_y = k_{N2O5} = 1 \times 10^{-3}$ s$^{-1}$ |

[Figure]

Figure S-5. Simulated temporal profiles of $NO_3$ and $N_2O_5$ (left axis) and $O_3$ and $NO_2$ (right axis). The subpanels A, B, and C are simulations with $k_{NO3} = 0$ $s^{-1}$, $1\times10^{-3}$ $s^{-1}$ or $1\times10^{-2}$ $s^{-1}$, respectively.

[Figure]

**Figure S-6**. Equilibrium constants for reaction (2) calculated for the three scenarios shown in
Figure S-5.

[Figure]

**Figure S-7**. Comparison of $\tau(N_2O_5)$ calculated using equation (2) of the main manuscript. with
the dashed lines calculated using equation (11) of Brown et al. (2003).

**Estimates of how loss of $NO_3$ to VOCs would affect the lifetime of $N_2O_5$**

The steady state lifetime calculation presented in Figure 8C of the main manuscript neglects losses of $NO_3$ to VOCs due to poor data coverage, i.e., presents a scenario where $\Sigma k_{NO3+VOC,i}[VOC]_i$ is assumed to be zero, which is, of course, unrealistic.

We used all available VOC data and calculated a time series of $\Sigma k_{NO3+VOC,i}[VOC]_i$. The average ($\pm 1\ \sigma$) value at night is $(0.038\pm0.026)$ s$^{-1}$. The $N_2O_5$ loss frequency, calculated by dividing this value with the $N_2O_5$:$NO_3$ ratio, is $(1.1\pm0.9)\times10^{-5}$ s$^{-1}$, corresponding to a lifetime of ~2.5 hours, which is negligible.

However, as stated in the main manuscript, the VOC data coverage is sparse and did not include measurements of all hydrocarbons towards which $NO_3$ is reactive. Recently, Liebmann et al. (2018) reported an average of nocturnal $NO_3$ loss frequency of 0.11 s$^{-1}$ in the boreal forest of Finland. This value likely included loss of $NO_3$ to NO and a variety of hydrocarbons such as sesqui- and diterpenes, which are likely present in higher concentration in a boreal forest than at Abbotsford and hence represents an upper limit. Taking this value and dividing it by the $N_2O_5$:$NO_3$ ratio, the average nocturnal $N_2O_5$ loss frequency via $NO_3$-VOC reactions is calculated to $(5.6\pm1.3)\times10^{-3}$ s$^{-1}$. Figure S-8 shows the result of including this value in the calculation of $N_2O_5$ lifetime.

[Figure]

**Figure S-8**. Same as Figure 8c but including an assumed $NO_3$ loss frequency to VOCs of 0.11 s$^{-1}$.